# The Cabrières Biota (France) provides insights into Ordovician polar ecosystems

Farid Saleh [1] ✉, Lorenzo Lustri[1], Pierre Gueriau [1,2], Gaëtan J.-M. Potin[1], Francesc Pérez-Peris[1,3], Lukáš Laibl[4], Valentin Jamart [1], Antoine Vite[1,5], Jonathan B. Antcliffe [1], Allison C. Daley[1], Martina Nohejlová[6], Christophe Dupichaud[5], Sebastian Schöder[7], Emilie Bérard[7], Sinéad Lynch [1], Harriet B. Drage [1], Romain Vaucher [1,8], Muriel Vidal[9], Eric Monceret[10], Sylvie Monceret[10] & Bertrand Lefebvre [5]

Early Palaeozoic sites with soft-tissue preservation are predominantly found in Cambrian rocks and tend to capture past tropical and temperate ecosystems. In this study, we describe the diversity and preservation of the Cabrières Biota, a newly discovered Early Ordovician Lagerstätte from Montagne Noire, southern France. The Cabrières Biota showcases a diverse polar assemblage of both biomineralized and soft-bodied organisms predominantly preserved in iron oxides. Echinoderms are extremely scarce, while sponges and algae are abundantly represented. Non-biomineralized arthropod fragments are also preserved, along with faunal elements reminiscent of Cambrian Burgess Shale-type ecosystems, such as armoured lobopodians. The taxonomic diversity observed in the Cabrières Biota mixes Early Ordovician Lagerstätten taxa with Cambrian forms. By potentially being the closest Lagerstätte to the South Pole, the Cabrières Biota probably served as a biotic refuge amid the high-water temperatures of the Early Ordovician, and shows comparable ecological structuring to modern polar communities.

Early Palaeozoic sites with soft-tissue preservation[1] provide a wealth of information on the evolution of past life and enhance our understanding of previous ecosystems[2,3], but are unequally distributed in time and space. While approximately 100 assemblages with soft-tissue preservation[4] have been described from the Cambrian, around 30 are known from the Ordovician[5–17], and only a few Lagerstätten are discovered in Early Ordovician rocks[4].

The distribution of Early Palaeozoic Lagerstätten is also palaeogeographically skewed, as approximately 97% of discovered biotas represent tropical and temperate ecosystems within 65° north and south of the palaeoequator[4]. This pattern is particularly true for the Ordovician, where very few Lagerstätten are known from polar environments[4]. Among the most famous Ordovician Lagerstätten, the Soom Shale (Upper Ordovician, South Africa), Big Hill (Late Ordovician, United States) and Winneshiek (Middle Ordovician, United States) biotas are indicative of tropical ecosystems[11–13] (Extended Data Fig. 1). The Liexi Fauna, along with the Fenxiang and Tonggao biotas from the Early Ordovician of China, represent tropical to warm temperate ecosystems[5–7] (Extended Data Fig. 1). The Afon Gam (Early Ordovician, United Kingdom), Castle Bank (Middle Ordovician, United Kingdom)

[1]Institute of Earth Sciences, University of Lausanne, Lausanne, Switzerland. [2]Université Paris-Saclay, CNRS, ministère de la Culture, UVSQ, MNHN, Institut photonique d'analyse non-destructive européen des matériaux anciens, Saint-Aubin, France. [3]Department of Earth and Environmental Sciences, University of Iowa, Iowa City, IA, USA. [4]Czech Academy of Sciences, Institute of Geology, Prague, Czech Republic. [5]Université de Lyon, Université Claude Bernard Lyon 1, École Normale Supérieure de Lyon, CNRS, UMR5276, LGL-TPE, Villeurbanne, France. [6]Czech Geological Survey, Prague, Czech Republic. [7]Synchrotron SOLEIL, L'Orme des merisiers, Gif-sur-Yvette, France. [8]Department of Earth Sciences, University of Geneva, Geneva, Switzerland. [9]Univ Brest, CNRS, Ifremer, Geo-Ocean, UMR 6538, Plouzané, France. [10]Société d'Etudes Scientifiques de l'Aude, Carcassonne, France. ✉e-mail: farid.nassim.saleh@gmail.com

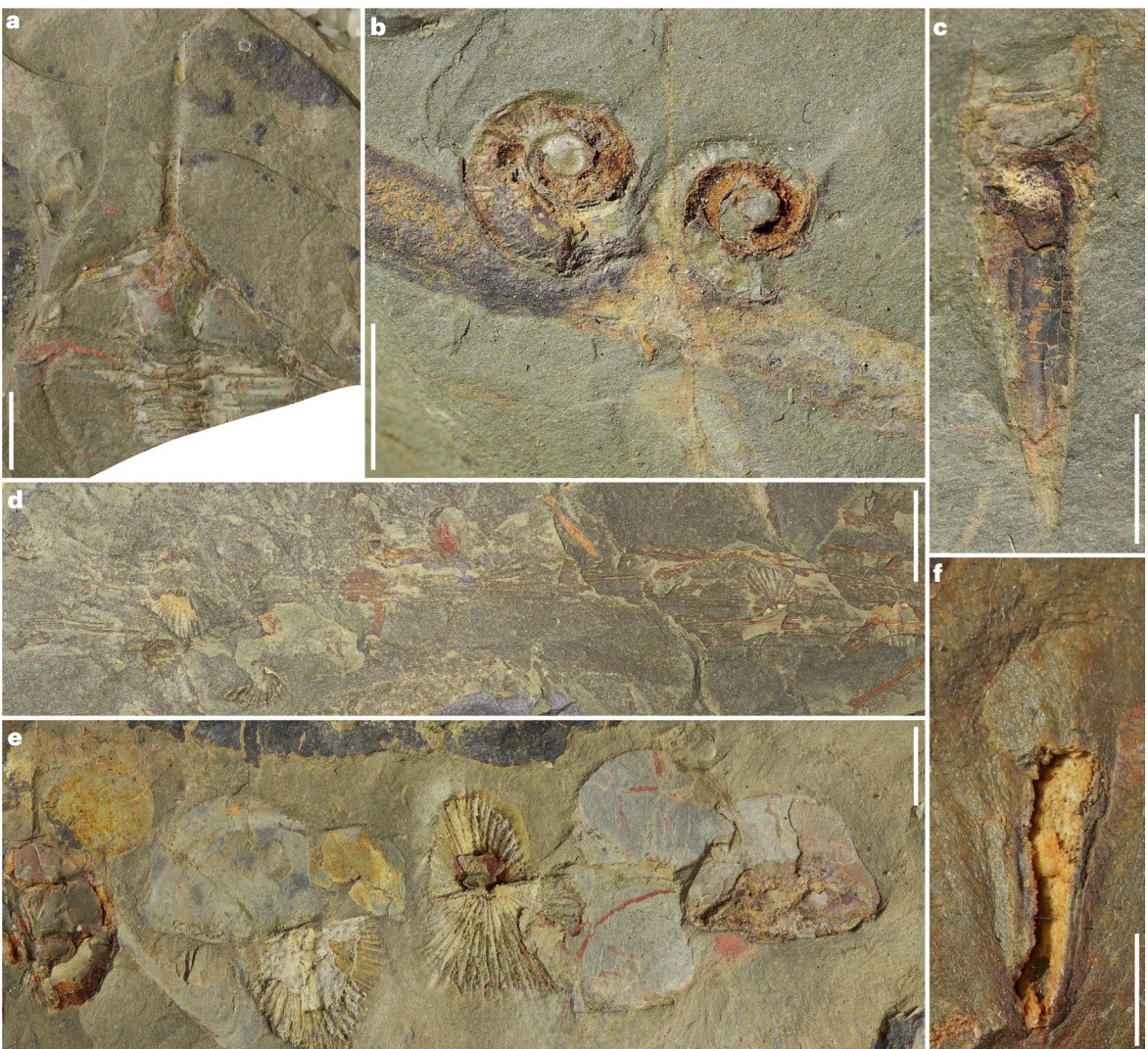

**Fig. 1 | Biomineralized taxa of the Cabrières Biota. a**, Trilobite of the genus *Ampyx* (UCBL-FSL713598). **b**, Gastropods associated with a tube-like structure, probably the conulariid *Sphenothallus* (UCBL-FSL713599). **c**, Biomineralized conulariid cnidarian (UCBL-FSL713600). **d**, Articulated brachiopods attached to a possible leptomitid sponge (UCBL-FSL713601). **e**, Assemblage formed of articulated brachiopods (centre), flattened carapaces probably of bivalved arthropods (centre left and right) and a calymenine trilobite cranidium (left; UCBL-FSL713602). **f**, A hyolith with possible internal organs (UCBL-FSL713603). Scale bars represent 4 mm in **a** and **e**, 1 cm in **b** and **d**, 5 mm in **c**, and 2 mm in **f**.

and Llanfawr (Middle Ordovician, United Kingdom) biotas provide valuable information on cold to temperate Ordovician communities near the polar circle[8–10] (Extended Data Fig. 1). The Early Ordovician Fezouata (Morocco) and Klabava (Czech Republic) biotas are the rare exceptions to this pattern, providing insights into strictly polar ecosystems[15,16] (Extended Data Fig. 1). Taken together, all these sites exhibit a mix of typical Cambrian and later Palaeozoic taxa, and suggest that marine assemblages were in transition between two early biodiversification events, the Cambrian Explosion and the Great Ordovician Biodiversification Event[18,19].

Considering the rarity of Ordovician Lagerstätten (Extended Data Fig. 1) and their skewed palaeogeographic distribution (Extended Data Fig. 1), the discovery of new biotas with soft-tissue preservation beyond the aforementioned palaeogeographic zones and environments is crucial for expanding our understanding of this time period and gaining better insights into the factors driving the rise of animal diversity on Earth. In this study, we describe a new fossil assemblage with soft-tissue preservation, the Cabrières Biota, from the Early Ordovician of southern Montagne Noire, France. The taxonomic diversity of this fossil biota is described, and the preservation of the fossils is investigated.

The recent findings are then discussed in light of other Early Ordovician Lagerstätten. This newly discovered biota is of particular importance as it is a close Ordovician Lagerstätte to the contemporaneous South Pole (Extended Data Fig. 1), constituting a cornerstone for understanding ancient polar ecosystems and their evolution.

## Results and discussion
### Stratigraphy and environmental context
The Early Ordovician Cabrières Biota is a newly discovered assemblage from the southern Montagne Noire, France (Extended Data Fig. 2). During the Early Ordovician, the Montagne Noire was an open marine environment located in the Southern Hemisphere at high polar latitudes on the margin of the supercontinent Gondwana[20] (Extended Data Fig. 1). The biota is preserved in stratigraphically equivalent layers to the Landeyran Formation, but more to the east than traditional localities, specifically within the *Apatokephalus incisus* trilobite biozone[21–24], which dates it to an upper Floian age[25] (Fl3; Extended Data Fig. 2). The Landeyran Formation corresponds to an offshore environment deposited in a transgressive phase[26,27], succeeding the sandy shoreface to upper offshore Foulon Formation[24,27]. However, a proper investigation

based on recent knowledge of mud deposition[28] is needed to properly frame the sedimentary context. Soft-tissue preservation occurs within an interval of 1 m thickness (Extended Data Fig. 2), located 15 m above the base of the Landeyran Formation (Extended Data Fig. 2).

## Faunal content

The biota contains numerous taxa that exhibit biomineralization (Fig. 1 and Extended Data Fig. 3). These include animals such as molluscs (14%), trilobites (12%), brachiopods (9%), hyoliths (7%) and cnidarians (6%) (Extended Data Fig. 3). Trilobites are primarily represented by the genera *Ampyx* (Fig. 1a), *Asaphellus* and calymenine trilobites (possibly *Colpocoryphe*) (Fig. 1e), which is in accordance with deposition in an open marine offshore environment[29]. Gastropods can be found in association with elongated tubes probably representing the enigmatic cnidarian (possible conulariid) *Sphenothallus* (Fig. 1b). Biomineralized conulariid cnidarians showing a quadrilateral aperture and phosphatized body are also preserved (Fig. 1c). Articulated brachiopods, mostly orthids, are abundant in the Cabrières Biota and can be observed either attached to possible leptomitid sponges with long monaxons[30] (Fig. 1d, and Extended Data Figs. 4 and 5) or randomly positioned near trilobites and possible non-biomineralized bivalved arthropods (Fig. 1e). Hyoliths are also present, although they are often poorly preserved. In one instance, a hyolith preserves possible traces of internal organs, probably representing the gut (Fig. 1f). An interesting feature of the Cabrières Biota is the rarity of echinoderms, which are represented by three specimens only (Extended Data Fig. 3).

In addition to trilobites, brachiopods, cnidarians, gastropods and hyoliths, the Cabrières Biota is characterized by a prevalence of sponges and branching algae constituting 26% of all identified fossils (Extended Data Fig. 3). Probable cylindrical demosponges can reach large sizes in excess of 10 cm. Specimen UCBL-FSL713604 is large with well-preserved subelliptical ostia within a thin dermal layer formed of fine fibres that are occasionally visibly spiculate (Fig. 2a and Extended Data Fig. 6). The termination of this fossil is unclear owing to incomplete preservation, obscuring whether it is a branched individual or two individuals close to each other (Fig. 2a). At the distal ends of the specimens, the oscula are not clearly defined (Fig. 2a). Other sponge specimens show detailed preservation (Fig. 2b), and the use of multispectral imaging allows for the differentiation of their soft tissues and skeleton (Fig. 2c). Algae of the Cabrières Biota vary in shape and size (Extended Data Fig. 7), including forms with a thick branching structure (Fig. 2d), delicate branching forms (Fig. 2e) and more intricate morphologies consisting of multiple compact branches and nodes (Fig. 2f). This site also preserves specimens (Fig. 2g) similar to *Margaretia* from the Burgess Shale, an organism previously attributed to green algae[31] but recently reinterpreted as organic tubes of the enteropneust hemichordate *Oesia*[32].

The Cabrières Biota also showcases a variety of bivalved arthropod carapaces forming 16% of identified fossils (Figs. 1e and 3a,b, and Extended Data Fig. 8). Most notable are the elongate suboval valves ornamented with very closely spaced, longitudinal striations (Fig. 3a,b), on rare occasions associated with abdominal segments not covered by the carapace (Fig. 3a), which represent a new taxon of phyllocarid crustacean. In addition, numerous fragments of non-biomineralized arthropods are present, including structures resembling chelicerate gnathobases (Fig. 3c) and a spiny appendage that could belong to either Radiodonta or Chelicerata (Fig. 3d). Some fossils of non-biomineralized arthropods exhibit segmented bodies adorned with ornamentation resembling that seen in chelicerates (Fig. 3e,f), sometimes with possible segmented appendages (Fig. 3f and Extended Data Fig. 9). One specimen also preserves what probably is a lunar-shaped eye and a rectangular prosoma (Fig. 3e), both of which are consistent with features seen in eurypterids, synziphosurids or even chasmataspids[33,34]. The post-prosomal anatomy of this specimen reveals a possible opisthosoma divided into a pre-abdomen and an abdomen (Fig. 3e and Extended Data Fig. 9).

Some vermiform organisms are also present in the Cabrières Biota (~1% of identified fossils), one of which exhibits external ornamentation consisting of many tiny nodes and preserves gut remains (Fig. 4a). Two other specimens consist of a partially preserved elongated and annulated soft body bearing two thick oval plates (Fig. 4b,c). These plates are approximately 2 mm and 6 mm long in the first and second specimens, respectively, and present a complex internal morphology (Fig. 4b,c) with an outer surface showing some reticulate ornamentation in places where thickness is preserved (Fig. 4d,e). A lateral extension at the base of one of the plates in the first specimen probably represents the remains of the proximal part of an appendage (?lo; Fig. 4b). At a similar position in the second specimen, a strong annulated area ends laterally into a series of lateral outgrowths (Fig. 4d,e) that probably represent spines or appendicules. The combination of a soft annulated body (and potentially appendages) and sclerite plates is characteristic of armoured lobopodians.

## Preservation mode

Fossils from the Cabrières Biota commonly exhibit brown, red or orange hues and are embedded within a siliciclastic matrix composed of mudstone and siltstone, which can range in colour from blue to green and yellow (Figs. 1–4). Scanning electron microscopy (SEM) backscattered electron and energy-dispersive X-ray (EDX) analyses indicate that the fossils are made of dense, shapeless iron oxide crystals lacking distinct framboids or euhedral minerals (Fig. 5a–c), surrounded by a matrix of aluminium-rich phyllosilicates (Fig. 5c). Synchrotron-based investigations of the chemical speciation of iron using Fe $K$-edge X-ray absorption near edge structure (XANES) spectroscopy show that iron is present as Fe(III) oxides and hydroxides (Fig. 5d; see Extended Data Fig. 10 for the position of the spectrum). In addition, black films, probably representing carbonaceous material, can be observed on some fossils (Fig. 5a).

The preservation of the Cabrières Biota exhibits similarities with the preservation seen in the Fezouata Biota, including comparable fossil colours and chemical signatures[35,36]. At least part of the iron oxides and hydroxides in the Cabrières Biota (Fig. 5a–d), such as the Fezouata Shale[36], may result from alteration by modern precipitation events because weathering products, such as manganese and arsenic, are deposited on the surface of the samples (Fig. 5e,f and Extended Data Fig. 10). The future collection of an expanded range of fossils will enable a more comprehensive taphonomic analysis of the modes and mechanisms of preservation within the Cabrières Biota and will facilitate comparisons with other Lagerstätten[37–41].

## Taxonomic and taphonomic importance

Many organisms of the Cabrières Biota are not fully mineralized and exhibit preservation of sclerotized, that is, toughened cuticle, in addition to cuticularized and cellular structures. As such, the Cabrières Biota is considered a Lagerstätte. Although the Montagne Noire region contains numerous fossil sites[24] with a wide range of temporal and palaeogeographical distributions, its Ordovician outcrops were previously recognized mainly for their biomineralized elements, such as trilobites[20–27,42–44], echinoderms[44–46], molluscs, brachiopods, hyoliths[21] and conulariids[47] as well as heavily sclerotized organisms such as graptolites[48]. The discovery of the Cabrières Biota expands the range of preserved tissue types found in the Ordovician of Montagne Noire, revealing entirely soft taxonomic groups such as algae and non-biomineralized animals (Figs. 2–4 and Extended Data Fig. 3).

Preliminary quantifications of the overall diversity within this biota reveal that organisms with biomineralized body walls (for example, brachiopods, echinoderms, trilobites) that do not preserve soft tissues make up approximately 41% of the total diversity. By contrast, over half of the total diversity comprises non-biomineralized organisms, such as bivalved arthropods, chelicerates, lobopodians and hemichordates, or biomineralized animal groups that do preserve soft tissues, such as the figured sponges (Fig. 2a,b). It is worth noting

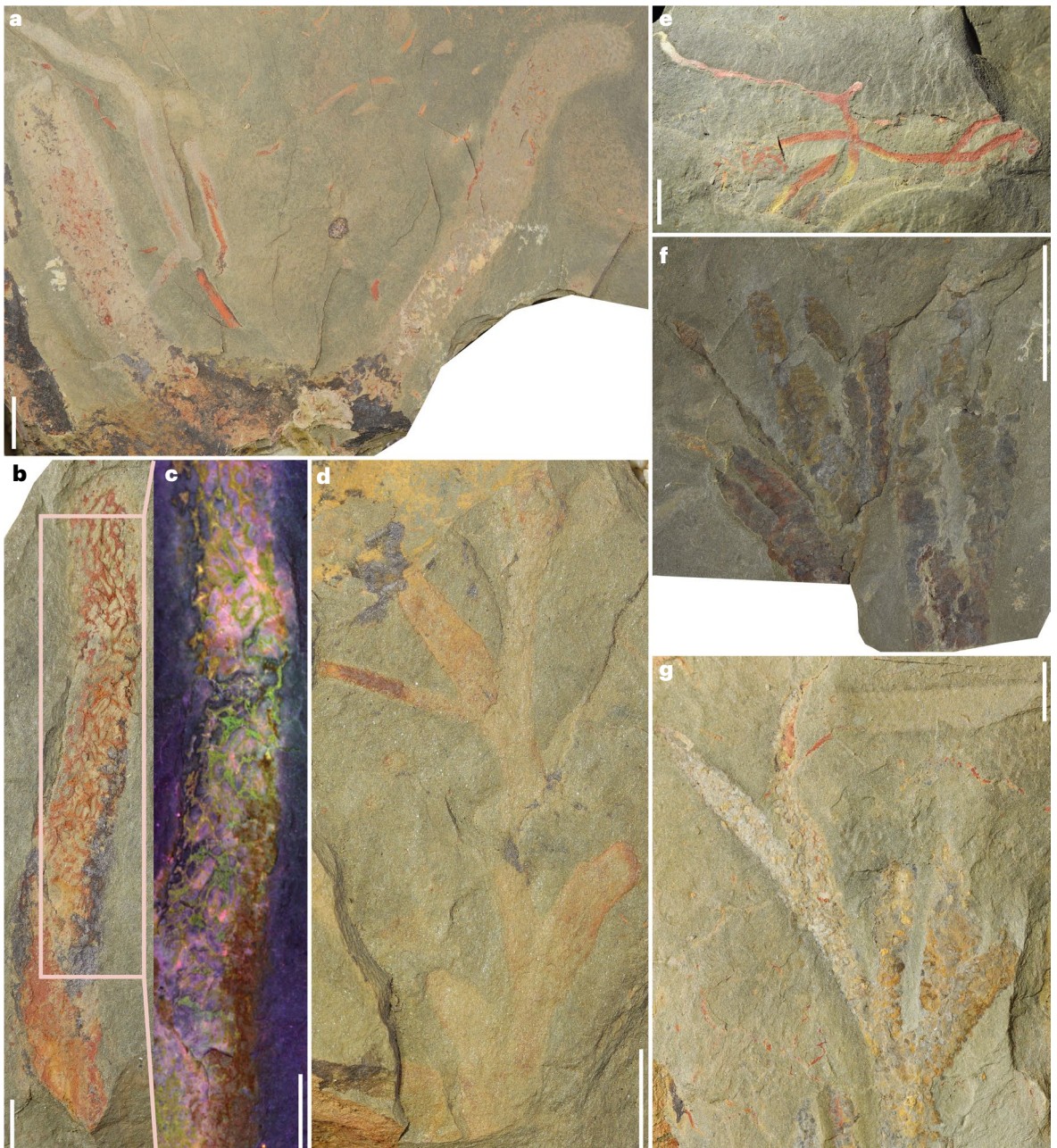

**Fig. 2 | Sponges, algae and possible hemichordates from the Cabrières Biota.**
**a**, A large sponge from the Cabrières Biota, possibly a demosponge (UCBL-FSL713604). **b**, Sponge (UCBL-FSL713605). **c**, Same specimen showing clear differentiation between the soft tissues (pink) and the mineralized skeleton (green) under multispectral imaging. **d**, Thick branching algae (UCBL-FSL713606). **e**, Thin branching algae (UCBL-FSL713607). **f**, More complex algae (UCBL-FSL713608). **g**, Organic tube of an *Oesia*-like enteropneust hemichordate (=*Margaretia*; UCBL-FSL713609). Scale bars represent 1 cm in **a** and **f**, 5 mm in **e**, and 3 mm in **b**–**d** and **g**.

that these percentages are comparable to those of other well-known Lagerstätten from the Early Ordovician, such as the Fezouata Biota, which has approximately 44% of its taxa[37,38], preserving only biomineralized remains. Moreover, many organisms in the Cabrières Biota can be fragmentary, which may indicate that they were either exposed to decay for relatively long periods of time or transported by sedimentary flows. Regardless of the processes responsible for such fragmentation, which will require further investigations, similar preservation is also observed in some localities from the Fezouata Biota, in which animals are dominantly fragmentary, with fully articulated organisms being the exception rather than the norm[49]. Despite the difference in collection efforts between the Fezouata Biota and the Cabrières Biota, which was only recently discovered, the latter still yielded some

complete organisms. Many animals are preserved in high detail as well, as exemplified by the longitudinal striations observed on the bivalved arthropod carapaces and the ornamentations on the chelicerates and the worms (Figs. 3a,b,e,f and 4, and Extended Data Figs. 8 and 9).

All animal groups in the Cabrières Biota are known from other Cambrian and Ordovician Lagerstätten, yet the taxonomic composition of the Cabrières Biota is particularly unique for the Early Ordovician. The newly described biota is almost as diverse as the range of clades seen in the Liexi Fauna[5] and Fezouata Biota[15], yet echinoderms, which are otherwise abundant in the Ordovician, are extremely rare in this biota. This scarcity of echinoderms in the Cabrières Biota is similar to that in the Fenxiang Biota[7] and the Klabava Biota[16] but differs from that of other Early Ordovician Lagerstätten such as the Leixi Fauna

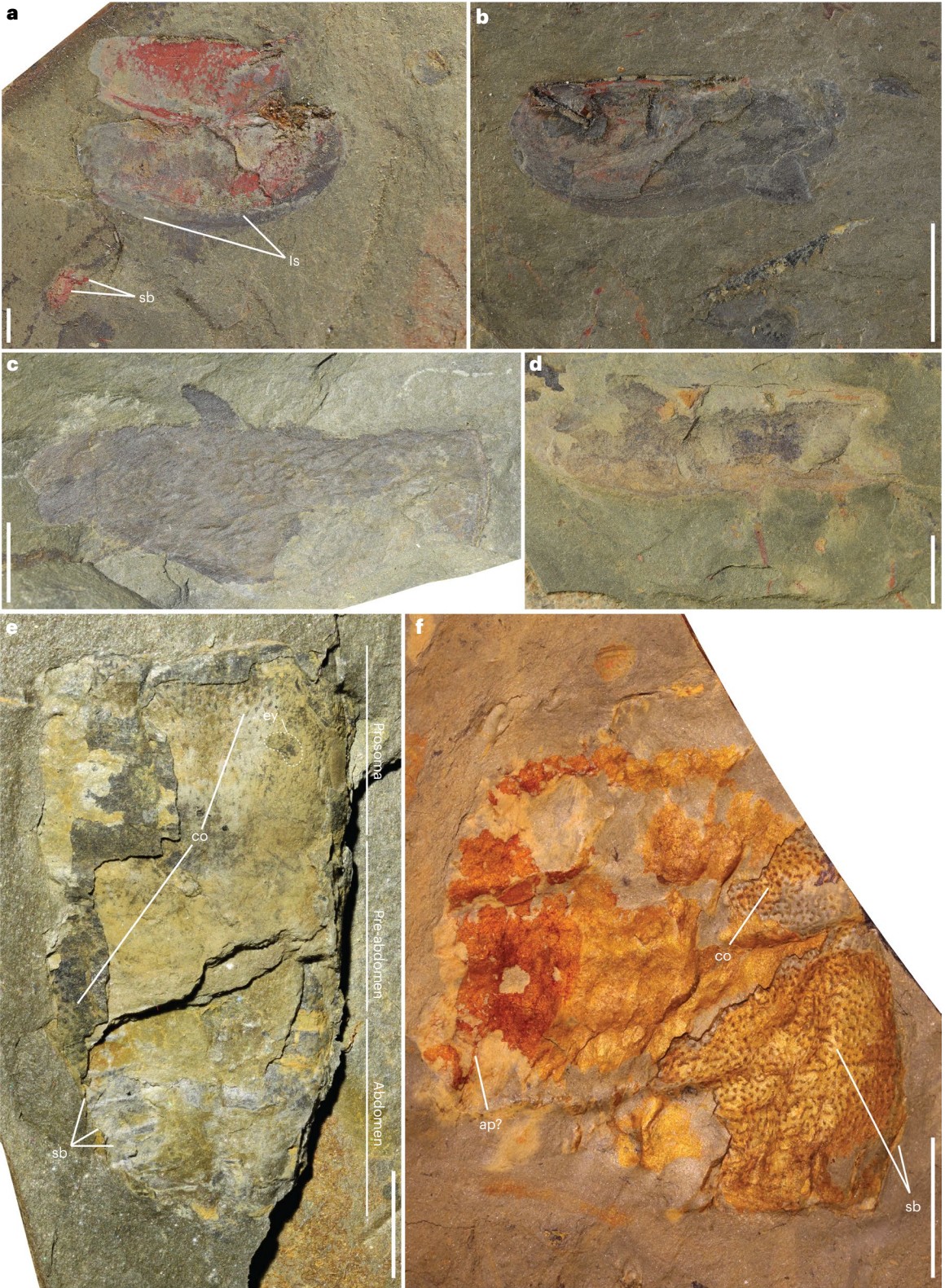

**Fig. 3 | Non-biomineralized arthropods of the Cabrières Biota. a**, Phyllocarid carapace valves ornamented with very closely spaced, longitudinal striations and associated with abdominal segments (UCBL-FSL713609). **b**, Phyllocarid carapace valve with longitudinal striations preserved near a graptolite (UCBL-FSL713610). **c**, Possible chelicerate gnathobase (UCBL-FSL713611). **d**, Spiny arthropod appendage (UCBL-FSL713612). **e**, Segmented arthropod with chelicerate-like ornamentation and an eye (UCBL-FSL713613). **f**, Part of a segmented arthropod with chelicerate-like ornamentation and an appendage (UCBL-FSL713614). ap, appendage; co, chelicerate ornamentation; ey, eye; ls, longitudinal striations; sb, segmented body. Scale bars represent 2 mm in **a**; 8 mm in **b**, **c** and **f**; 5 mm in **d** and **e**; and 4 mm in **j**.

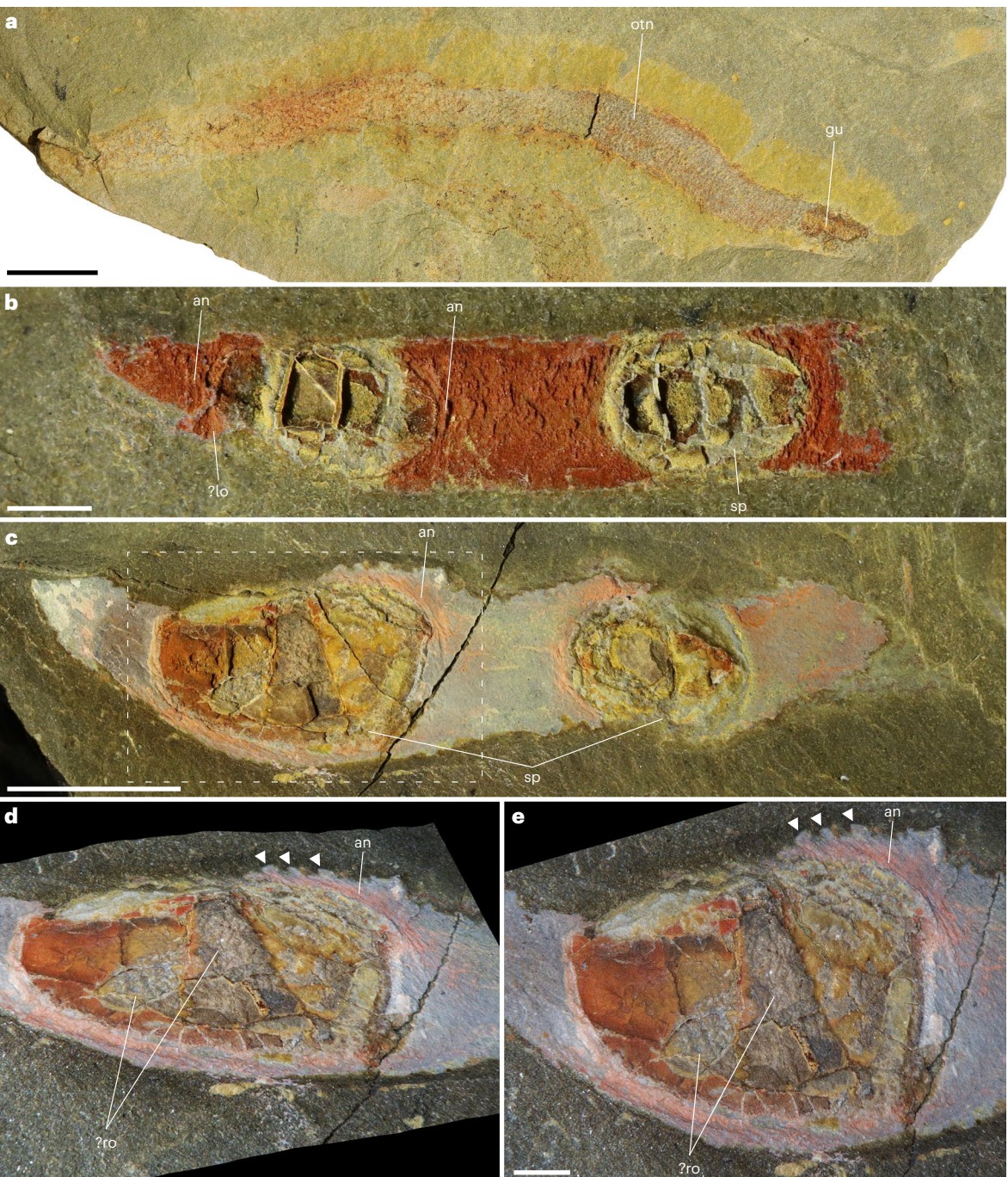

**Fig. 4 | Vermiform organisms from the Cabrières Biota. a**, Unidentified vermiform organism UCBL-FSL713615, with possible remains of the gut and external ornamentation of tiny nodes. **b**,**c**, Incomplete armoured lobopodians UCBL-FSL713616 (**b**) and UCBL-FSL713617 (**c**) exhibiting two sclerite plates along an elongated soft body with annulations. A lateral extension in **b** possibly represents remains of the proximal part of a lobopod (?lo). **d**,**e** Close-up three-dimensional lateral (**d**) and top (**e**) views of part of UCBL-FSL713617, from the dotted box in **c**. Arrowheads point to lateral outgrowths associated with annulations that could represent spines or lobopod appendicules. an, annulations; gu, gut; lo, lobopod; otn, ornamentation of tiny nodes; ro, reticulate ornamentation; sp, sclerite plates. Scale bars represent 5 mm in **a** and **c**, and 1 mm in **b** and **e**; note that due to the three-dimensional rotation, no scale bar is given for **d** and the reader is instead invited to refer to scale bars in **c** and **e**.

and particularly the Fezouata Shale[50–56] (Extended Data Fig. 3). The Cabrières Biota yields a higher diversity of arthropods compared with the Fenxiang Biota and lacks evidence of nematodes, scalidophorans and corals. Furthermore, there are no bryozoans present in the Cabrières Biota in contrast to the Klabava Biota. The Cabrières Biota preserves an abundance of algae and sponges (Fig. 2a–f and Extended Data Fig. 3), similar to the Afon Gam Biota[9], but with a greater number of non-biomineralized arthropods (Fig. 3a–f). The Cabrières Biota provides further evidence that armoured lobopodians (Fig. 4b,c) persisted until at least the Ordovician. Armoured lobopodians were important components of Cambrian ecosystems and are highly abundant in Cambrian Lagerstätten, such as the Chengjiang Biota[57], and can be present in Ordovician ecosystems[58]. With the discovery of the Cabrières Biota, it is becoming clearer that many elements of the classic Cambrian fauna persisted into the Ordovician. Findings such as these are increasingly connecting the Cambrian Explosion and the

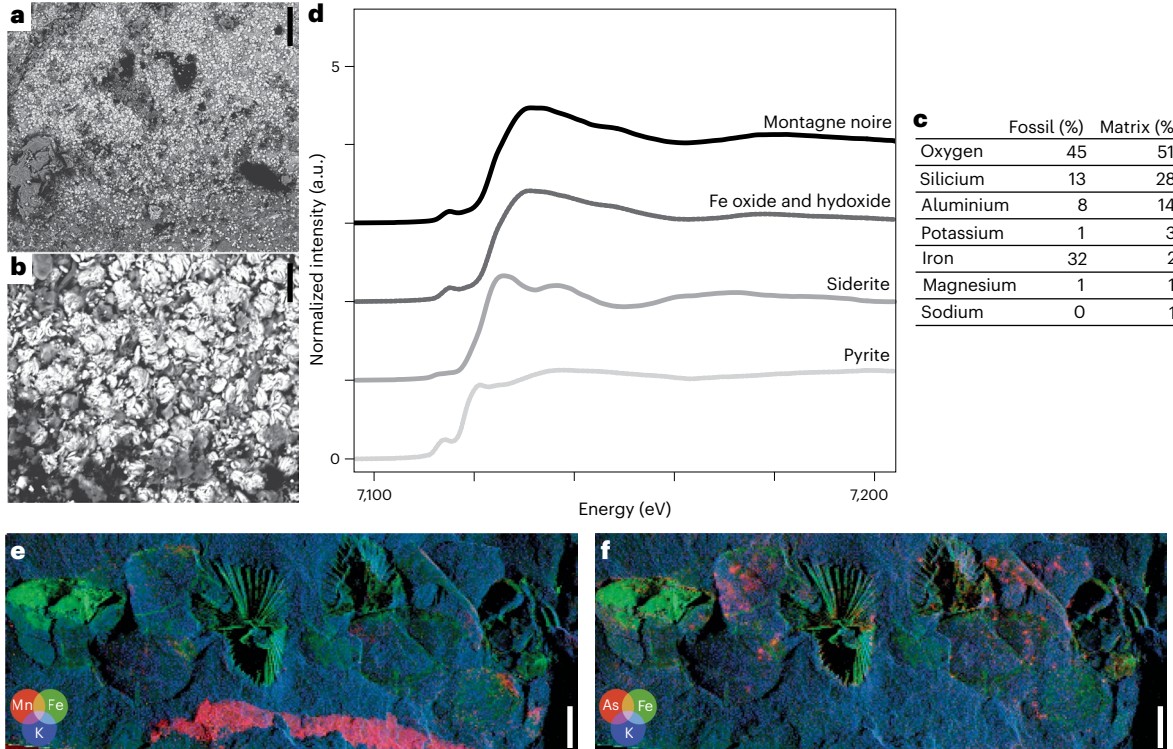

| **c** | Fossil (%) | Matrix (%) |
|---|---|---|
| Oxygen | 45 | 51 |
| Silicium | 13 | 28 |
| Aluminium | 8 | 14 |
| Potassium | 1 | 3 |
| Iron | 32 | 2 |
| Magnesium | 1 | 1 |
| Sodium | 0 | 1 |

**Fig. 5 | Mode of fossil preservation. a,** Backscattered electron microscopy image revealing white iron oxide minerals and some limited black carbonaceous material within the fossils of the Cabrières Biota. **b,** The iron oxides appear shapeless, lacking distinct framboids or euhedral minerals. **c,** Semi-quantitative elemental proportions from SEM–EDX analyses indicate that the fossils exhibit a higher iron content compared with their surrounding aluminosilicate matrices. **d,** Fe K-edge XANES spectroscopy shows that the iron present in the Montagne Noire fossils exists in the form of oxides and hydroxides. **e,f,** Synchrotron μXRF major-to-trace elemental mapping shows that modern weathering elements such as manganese and arsenic are deposited on the surface of the samples. Scale bars represent 100 μm in **a,** 50 μm in **b** and 5 mm in **e** and **f.**

Great Ordovician Biodiversification Event[59] albeit existing differences between them[60].

## Ecological and evolutionary implications

The Cabrières Biota represents a close Lagerstätte to the Ordovician South Pole (Extended Data Fig. 1). High-latitude marine habitats can offer a range of favourable spatiotemporal conditions supporting high species richness[61], and play a crucial role as taxonomic refugia during periods of environmental stress[62–64]. Given the extremely warm waters of the Early Ordovician[65], the high-latitude Cabrières Biota would have experienced less extreme temperatures compared with lower-latitude regions, fostering the development of a unique diversity of taxa that had shifted southwards into cooler climatic bands. For instance, trilobite fauna from Montagne Noire shares taxa with Iran such as *Taihungshania* and *Damghanampyx*. The latter genus is only known in these two regions, while *Taihunghsania* also occurs in Turkey, United Arab Emirates and South China[66–69]. The Montagne Noire also shares numerous taxa with the Anti-Atlas in Morocco[70–73]. This melting pot in the Montagne Noire is restricted to the Lower Ordovician, and faunal affinities become strictly Gondwanan during the Upper Ordovician[74,75]. The unique taxonomic diversity of the Montagne Noire during the Early Ordovician might have been facilitated by the oceanic circulation permitting taxa to migrate towards the pole from warmer, more stressful, lower latitudes[66], and might have been made easy by a possible position of the Montagne Noire in a slightly more eastern location within the polar circle[76–78].

Some modern polar biotas, similar to the Cabrières Biota, can be dominated by algae and sponges[79–82]. Macroalgae possess specific characteristics and adaptations in polar settings, which explain their ecological success in these environments[83]. Sponges play a key role in the community dynamics of polar settings[84] and can attain large sizes[85], as was observed in the Cabrières Biota. The prevalence of sponges in the Cabrières Biota cannot be ascribed to environmental factors such as oxygen depletion, even though sponges typically thrive in hypoxic environments. This is because hypoxic environments are characterized by low diversity, which is clearly not the case for the Cabrières Biota preserving a diverse array of organisms, including brachiopods, trilobites, bivalved arthropods, lobopodians, worms, cnidarians, hyoliths and molluscs. The diversity of arthropods in polar ecosystems is comparable to that of less harsh environments in the sub-Arctic and low-Arctic regions[86], which could explain the similarities in general arthropod diversity between the Cabrières Biota and nearby Lagerstätten, such as the Fezouata Biota (that is, the presence of trilobites, bivalved arthropods, chelicerates and possibly radiodonts)[87–89]. Cnidarians are also present in modern polar ecosystems[90], as is the case in the Cabrières Biota. Their success is related to their wide range of diets and opportunistic behaviour, enabling them to take advantage of the available food sources in these extreme ecosystems[90].

The patterns for echinoderms are more complex. Echinoderms can be found in the Arctic and on the Antarctic margins[91,92]. However, their diversity is lower in these regions compared with other areas such as the Atlantic or Indo-Pacific oceans[93]. Within a certain polar setting, echinoderms can be locally abundant[91,92]. The scarcity of echinoderms in the Cabrières Biota could be attributed to its polar settings, among other local factors, especially that other sites from the Early Ordovician of Montagne Noire[45] have yielded diverse assemblages of echinoderms. Thus, similarly to modern polar ecosystems, echinoderms did not colonize the entire seafloor in the Montagne Noire. In fact, echinoderms thrive when the diversity of other animal groups is limited[52]. By adapting to oligotrophic conditions, echinoderms are

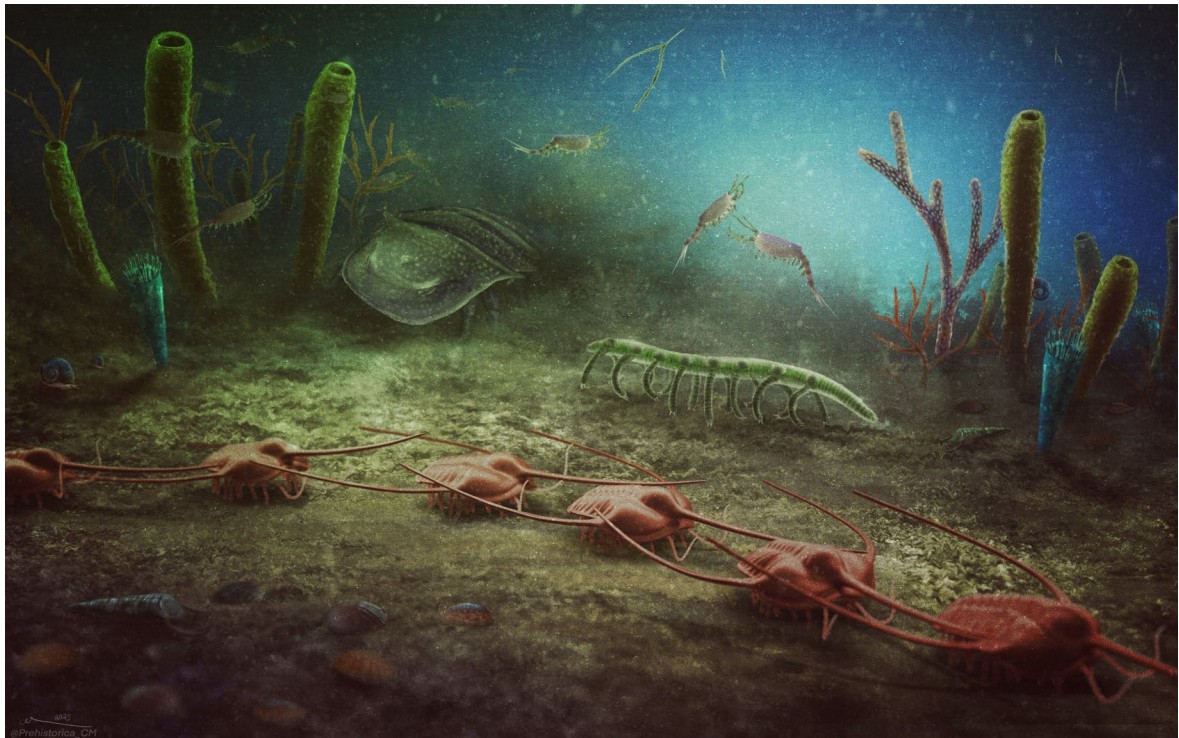

**Fig. 6 | Artistic reconstruction of the Cabrières Biota.** In the foreground, a row of *Ampyx* (trilobites) and various shelly organisms, including brachiopods and a hyolith (bottom left corner). Behind the trilobites, a lobopodian, a chelicerate, cnidarians (blue), sponges (green), thin branching algae (red and green) and hemichordate tubes (purple), along with some molluscs. Bivalved arthropods inhabit the water column along with graptolites. Credit: Christian McCall (Prehistorica Art).

often found in low-diversity, low-competition assemblages, where the conditions of the water column did not allow for the colonization of other animal groups, a pattern that is also respected in the Fezouata Biota. For instance, when examined at high resolution, echinoderms are abundantly found in levels of specific bathymetry where other animals are not diverse, constituting the building blocks for the high echinoderm diversity in the Fezouata Biota[49]. It is likely that echinoderms are rarely found in the Cabrières Biota owing to its high diversity and therefore increased competition, unlike nearby environments from the Montagne Noire that were favourable for echinoderm colonization. This pattern would have been accentuated if the bathymetric conditions in the Cabrières Biota were unfavourable for echinoderm colonization. This would explain echinoderm abundance in traditional Montagne Noire localities and their absence in the specific level yielding the Cabrières Biota.

The position of the Cabrières Biota in the polar zone, the preservation of a diverse assemblage, the dominance of sponges and algae, and the habitat selectivity of echinoderms, among other observed patterns such as the absence of bryozoans that are dominantly found in tropical and temperate waters, all support the notion that this assemblage represents a polar ecosystem characteristic of the Early Ordovician. The resemblance between the Cabrières Biota (Fig. 6) and modern polar ecosystems raises questions on how warmer Early Ordovician polar regions[65] supported similar niche partitioning and structuring to colder modern-world polar ecosystems. Moreover, considering the global cooling happening after the Early Ordovician, similar ecosystems to the high-latitude Cabrières Biota can be anticipated in low-latitude environments during Middle to Late Ordovician. Further global-scale investigations are required to confirm this trend, despite the presence of local palaeontological evidence suggesting the existence of low-latitude sponge-dominated Lagerstätten in the Late Ordovician of China[94]. Yet, it is possible to suggest that refugial zones found at high latitudes in the Early Ordovician migrated to lower latitudes during the subsequent Ordovician cooling.

## Methods

The search for Lagerstätten in the Early Ordovician of Montagne Noire (France) has gained momentum over the past decade. In 2018, the first potential soft tissues were discovered, and new discoveries by two authors (E.M. and S.M.), from the 'écailles de Cabrières', followed since. The fossiliferous sites are found in outcrops within a 1 km radius of the Cabrières village (Extended Data Figs. 1 and 2). Over 400 fossils have been collected so far, and all are registered and housed under the collection 'Monceret' at the University of Lyon (Université Lyon 1, Villeurbanne, France), under the acronym UCBL-FSL.

Collected fossils were examined with a WILD type 308700 (×6.4, ×16 and ×40) binocular microscope. Optical photos were taken with a Canon 800D camera coupled to a Canon MP-E 65 mm macro lens equipped with a polarizing filter. Various lighting conditions, including normal and polarized light, as well as dry and alcohol-submerged photography, were used. Z-stacks were processed using Helicon Focus software and the pyramid function. Three-dimensional surface images were produced for one specimen through an automatic vertical stacking process using a Keyence VHX-7000 digital microscope equipped with a VH-ZOOT Macro lens (×0–50 magnification) connected to a VXH-7020 high-performance 3.19-megapixel complementary metal-oxide semiconductor (CMOS) camera.

Some specimens were further documented using multispectral imaging at the Institute of Earth Science of the University of Lausanne (Switzerland) to see whether certain anatomies are better seen under different light combinations. Reflection and luminescence images in various spectral ranges were collected using a set-up consisting of a low-noise 2.58-megapixel back-illuminated sCMOS camera with high sensitivity from 200 to 1,000 nm, fitted with a UV–VIS–IR 60 mm 1:4 Apo Macro lens (CoastalOptics) in front of which is positioned a filter wheel holding eight interference band-pass filters (Semrock) to collect images in eight spectral ranges from 435 to 935 nm. Illumination was provided by 16 light-emitting diodes, with wavelengths ranging

from 365 nm to 700 nm (CoolLED pE-4000), coupled to a liquid light guide fitted with a fibre-optic ring light guide. As such, more than 90 different illumination and detection couples are available, and the resulting greyscale images can be combined into false-colour RGB images to enhance morphological contrasts or reveal details invisible in traditional photography using only visible light. Stacking, image registration of the different couples (excitation/emission of 385/935, 660/775 and 365/571) and production of false-colour RGB composites were performed using ImageJ.

Four specimens were analysed using an FEI Quanta 250 SEM at the Electron Microscopy Facility of the Faculty of Biology and Medicine of the University of Lausanne to investigate the mode of preservation of the Cabrières Biota fossils and their mineralogical composition. The SEM was equipped with backscattered and secondary electron detectors in addition to an EDX analyser. To detect heavy elements such as iron, specimens were analysed uncoated in environmental mode using a 10 keV beam. Elemental percentages (semi-quantifications) were obtained using the associated Bruker Nano Analytics software.

The trace elemental composition of two fossils was further investigated using synchrotron micro-X-ray fluorescence (μXRF) mapping at the PUMA beamline of the SOLEIL synchrotron source (France) to better constrain the differences between matrices and fossils. The incoming monochromatic X-ray beam was focused using Kirkpatrick–Baez mirrors down to a spot size of $\sim 7 \times 5\ \mu m^2$ (Horizontal × Vertical, full width at half maximum) and set to an energy of 18,500 eV optimized for the excitation of elements from phosphorus to zirconium (K-lines) and from cadmium to uranium (L-lines). The sample was mounted on a scanner stage allowing 150 mm and 100 mm movements (in horizontal and vertical directions, respectively) with micrometre accuracy, and oriented at 60° to the incident beam, producing an effective beam size of $\sim 10 \times 5\ \mu m^2$ (Horizontal × Vertical) on the sample. XRF data were collected using a SiriusSD silicon drift detector (SGX Sensortech, 100 mm² active area) oriented at 90° to the incident beam, in the horizontal plane. Two-dimensional spectral images, that is, images for which each pixel is characterized by a full XRF spectrum, were collected with a 60–80 ms dwell time at a 100–200 μm lateral resolution depending on the samples (see figure captions for the precise scanning parameters). The results are shown herein as false-colour RGB overlays of three elemental distributions reconstructed from full spectral decomposition using the batch-fitting procedure of the PyMCA data-analysis software[95], with polynomial baseline subtraction, and assuming a Hypermet peak shape. XANES spectroscopy at the Fe K-edge was performed to determine iron speciation. Fe XANES spectra were collected in fluorescence mode in the 7,050–7,550 eV range with energy steps of 5 eV between 7,050 and 7,100 eV, 0.5 eV between 7,100 and 7,200 eV, and 2 eV between 7,200 and 7,250 eV. The count time was set to 2 s per energy step. Energy was calibrated using a reference metallic Fe foil and setting the first inflection point of the Fe K-edge at 7,111 eV.

### Reporting summary

Further information on research design is available in the Nature Portfolio Reporting Summary linked to this article.

## Data availability

All the data needed to reproduce this paper are available in the main text and the extended data figures. All material can be accessed at the public collections of the University of Lyon (France).

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

## Acknowledgements

We thank A. Mucciolo and R. Abi Habib for their assistance during SEM analyses. We also thank D. Vizcaíno, C. Goujon and M. Goujon for their assistance during fieldwork. We are grateful to SOLEIL Synchrotron for the provision of beamtime and to A. Logghe and S. Sanchez for help at the PUMA beamline. We also thank L. Parry for participating in discussions during his visit to the University of Lausanne. This paper is a contribution to the International Geoscience Programme (IGCP) Project 735—Rocks and the Rise of Ordovician Life (Rocks n'ROL). F.S. thanks the Faculty of Geoscience and Environment of the University of Lausanne and the Swiss National Science Foundation (SNF; Ambizione grant, number PZ00P2_209102) for funding. L. Laibl was supported by the programme Dynamic Planet Earth of the Czech Academy of Sciences (StrategieAV21/30) and by the institutional support RVO 67985831 of the Institute of Geology of the Czech Academy of Sciences. M.N. was supported by an internal grant from the Czech Geological Survey (number 311410), part of the Strategic Research Plan of the Czech Geological Survey (DKRVO/CGS 2023–2027). H.B.D., S.L. and J.B.A. are funded by an SNF Sinergia grant (number CRSII5_198691) awarded to A.C.D. and three other principal investigators. L. Lustri and F.P.-P. were funded during the early stages of this project through an SNF project grant (number 205321_179084) awarded to A.C.D. V.J. is funded by an SNF Ambizione grant (PZ00P2_193520). G.J.-M.P. is supported by the Canton de Vaud.

B.L. is funded by the project ECO-BOOST of the French National Research Agency (ANR-22-CE01-0003).

## Author contributions

F.S., B.L. and M.N. did fieldwork in 2018 upon invitation from E.M., and E.M. and S.M. found the fossils after extensive fieldwork over 3 years, did the initial fossil identification and donated the samples to the University of Lyon. F.P.-P., G.J.-M.P., L. Laibl and V.J. helped with the inventory of the collection. Fossil examination included all authors, particularly L. Lustri and P.G. F.P.-P., L. Lustri, S.L., V.J. and G.J.-M.P. took photographs of the fossils. F.S. examined specimens using electron microscopy, A.V. applied multispectral imaging on selected samples, and P.G., S.S. and E.B. carried out synchrotron analyses. All authors interpreted and discussed the results. F.S. wrote the paper and made the figures with the help of all co-authors.

## Funding

## Competing interests

The authors declare no competing interests.

## Additional information

**Extended data** is available for this paper at https://doi.org/10.1038/s41559-024-02331-w.

**Correspondence and requests for materials** should be addressed to Farid Saleh.

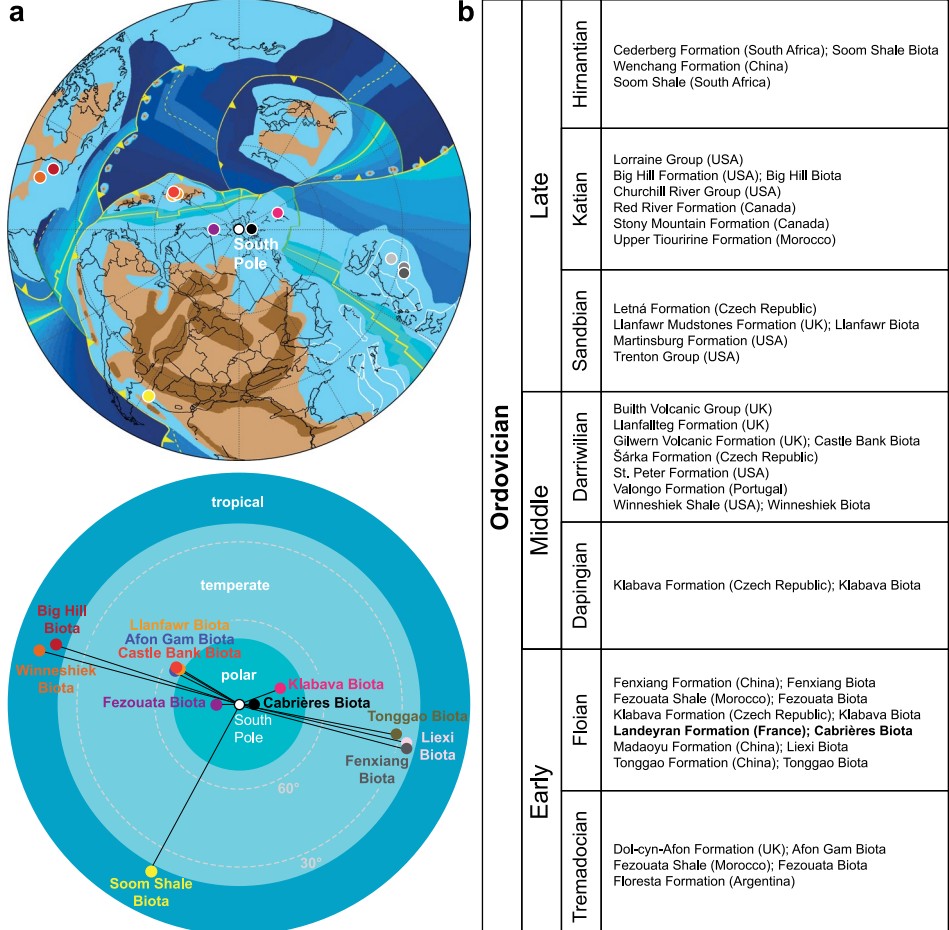

**Extended Data Fig. 1 | Distribution of Ordovician Lagerstätten. (a)** Early Ordovician palaeomap showing some of the significant Ordovician Lagerstätten, in addition to the newly discovered Cabrières Biota. **(b)** Stratigraphic formations yielding soft-tissue preservation during the Ordovician. The Cabrières Biota from the Landeyran Formation is the close to the Ordovician South Pole.

It should be noted that some of the plotted sites in (a) date from the Middle or Upper Ordovician and not from the Early Ordovician. The paleomap is modified from Scotese[96] but different reconstructions[97] show rather comparable paleogeographic distributions. Dapingien (Dapin), Hirnantian (Hirn).

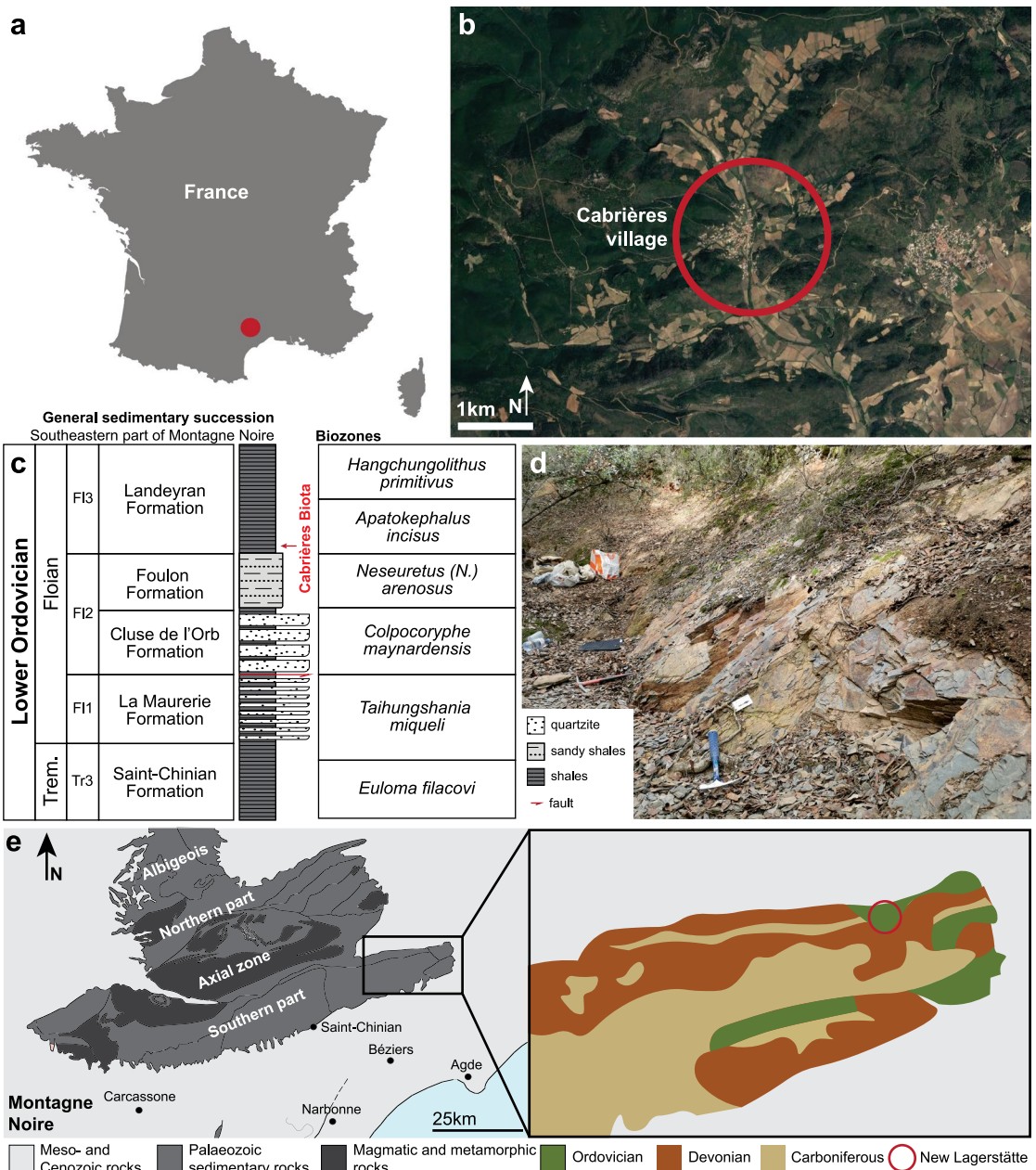

**Extended Data Fig. 2 | Geographic and stratigraphic context.** (**a**) Geographic position of the Cabrières Biota in the Montagne Noire in southern France, (**b**) within a 1 km radius around the Cabrières village. Map data ©2022 Google. The Cabrières Biota is discovered in the late Floian (Fl3) Landeyran Formation (**c**). (**d**) Photograph of the outcrop that yielded most soft-tissue preservation. (**e**) Geological map of the Montagne Noire showing the position of the new Lagerstätte in the southern part of the complex.

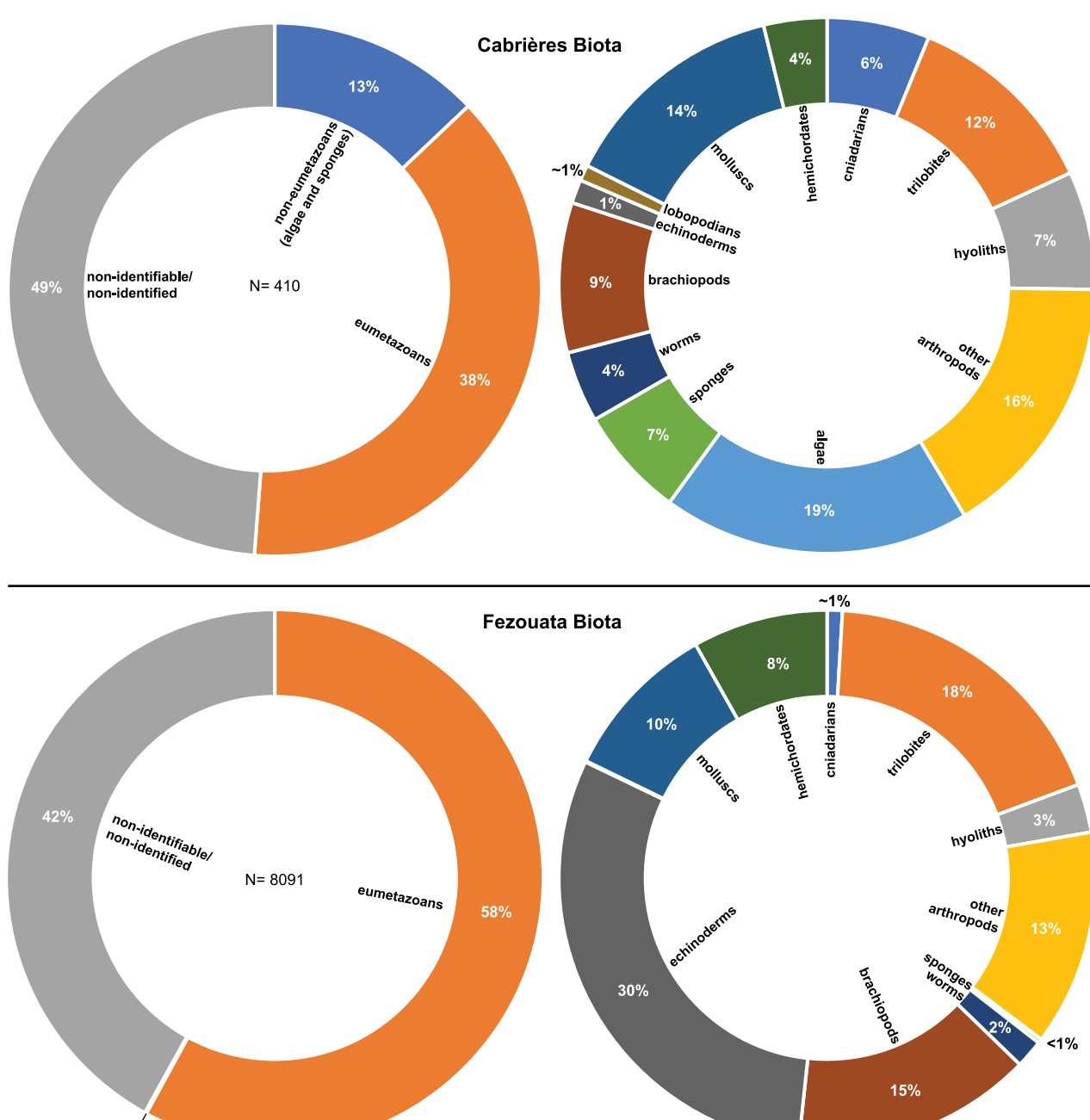

**Extended Data Fig. 3 | Taxonomic abundances in the Cabrières Biota and their comparison with the Early Ordovician Fezouata Biota.** The Cabrières Biota showcases a notably higher abundance of algae and sponges, while echinoderms are remarkably scarcer in comparison to the Fezouata Biota. Other animal taxa display relatively comparable representation between the two Lagerstätten. It is important to note that this data is highly probable to evolve with forthcoming fossil discoveries. Hence, it should be used only as an initial comparative reference. The Fezouata Biota dataset derives from the Marrakesh collections.

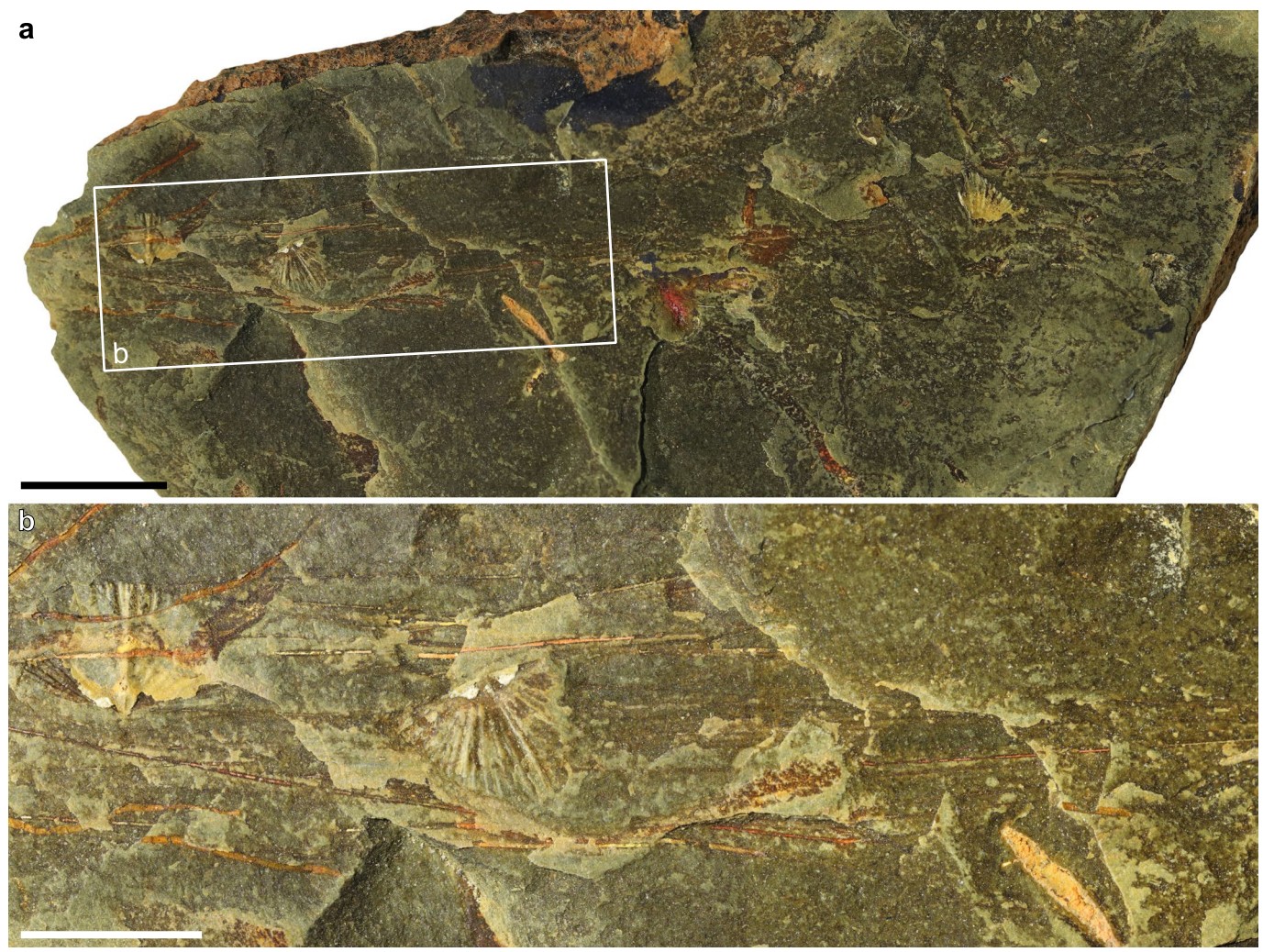

**Extended Data Fig. 4 | Additional views of the possible leptomitid sponge UCBL-FSL713601 shown in Fig. 1d.** (**a**) Optical photograph under polarised light of the entire specimen. (**b**) Close-up view under polarised light from the box in (a) showing projected longitudinal spicules. Scale bars represent 1 cm in (a), and 5 mm in (b).

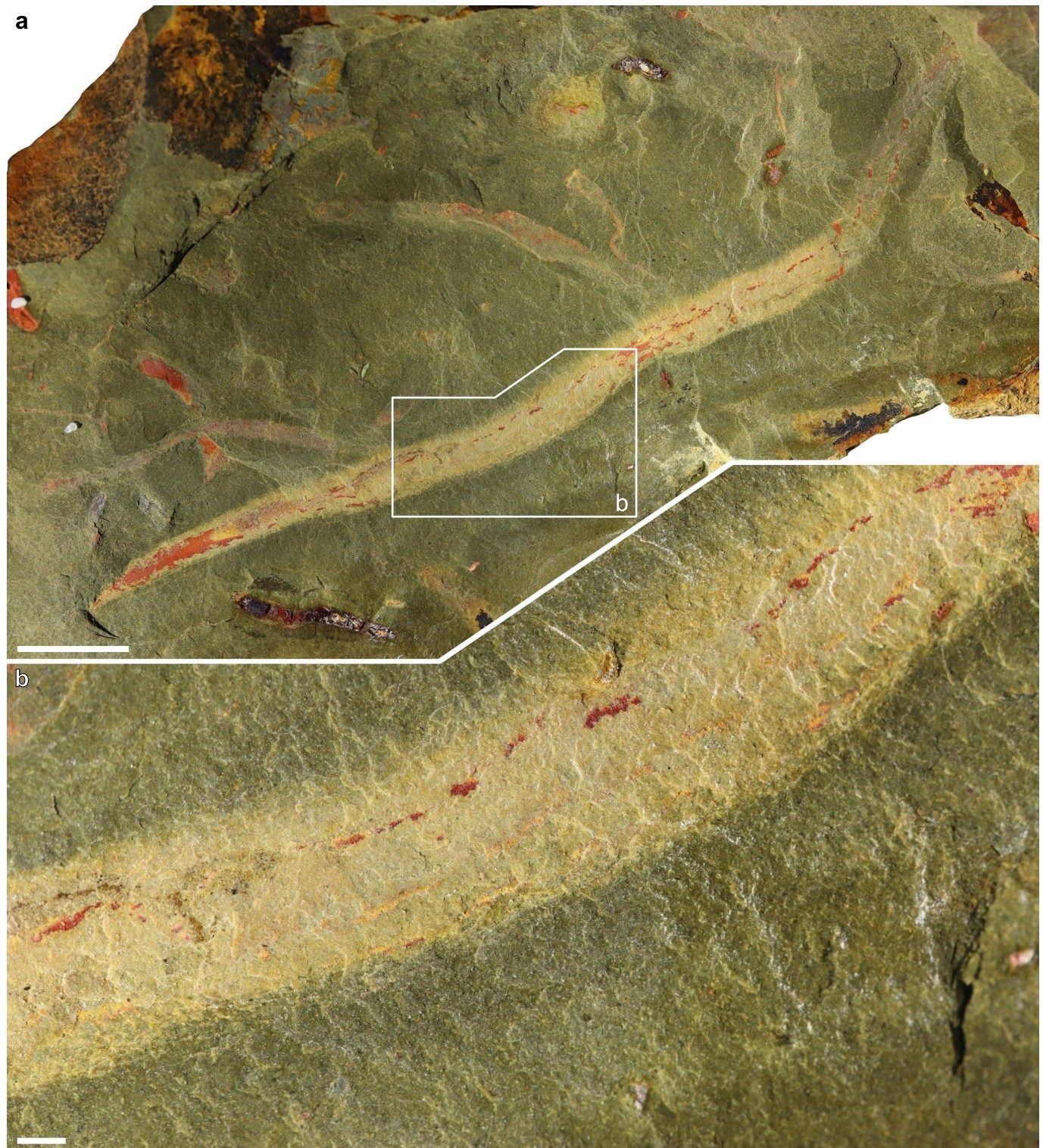

**Extended Data Fig. 5 | Another possible leptomitid sponge, associated to the thin branching algae UCBL-FSL713607. (a)** Optical photograph under polarised light of the entire specimen. (**b**) Close-up view under polarised light from the box in (a) showing longitudinal skeletal elements. Scale bars represent 1 cm in (a), and 1 mm in (b).

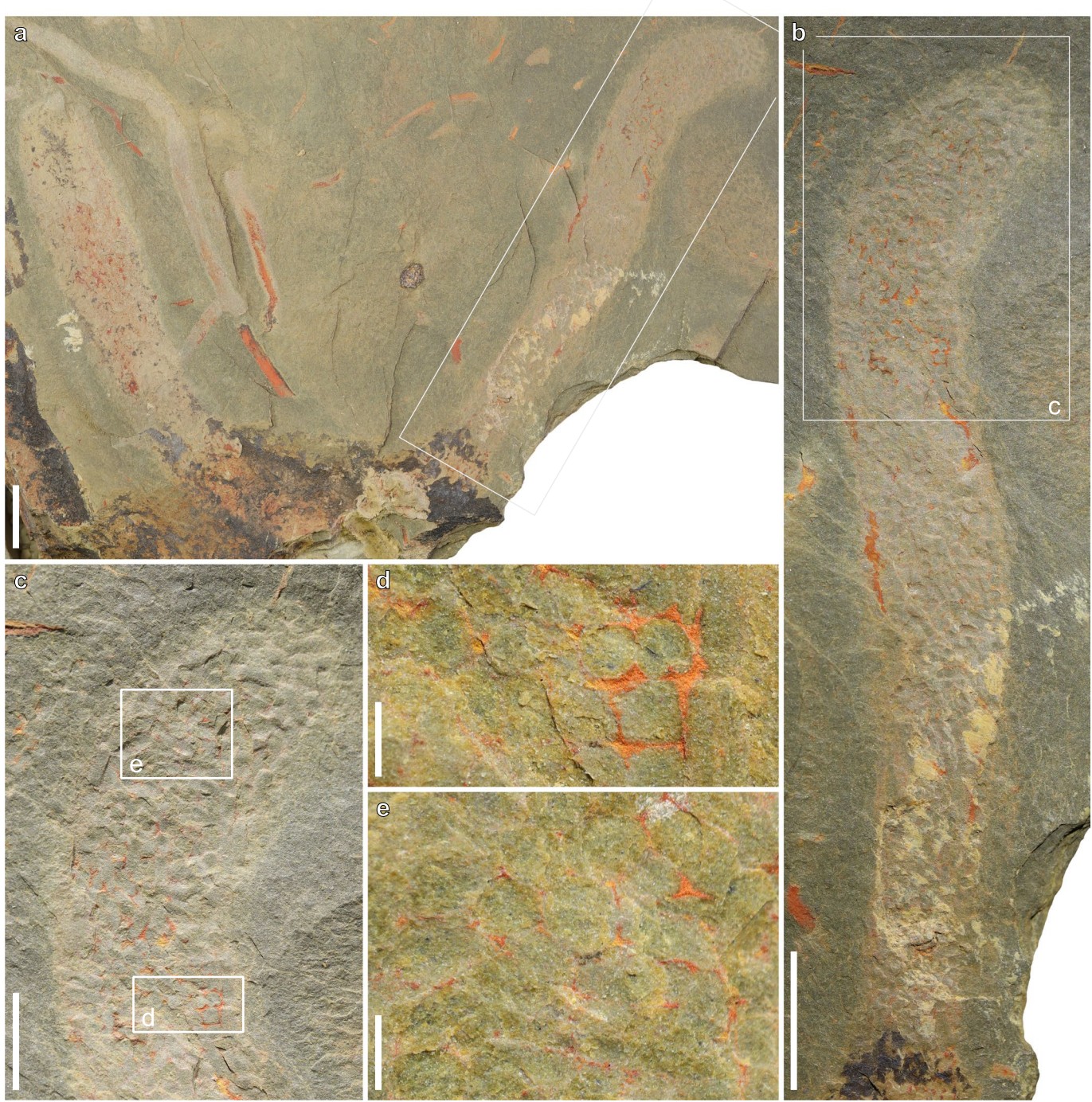

**Extended Data Fig. 6 | Additional views of the large sponge UCBL-FSL713604 shown in Fig. 2a.** (**a**) Optical photograph of the entire organism. (**b**) Close-up view of one side of the organism, from the box in (a). (**c**) Close-up view of the osculum from the box in (b). (**d**, **e**) Close-up views under polarised light showing the skeletal framework, from the corresponding boxes in (c). Scale bars represent 1 cm in (a) and (b), 5 mm in (c), and 1 mm in (d) and (e).

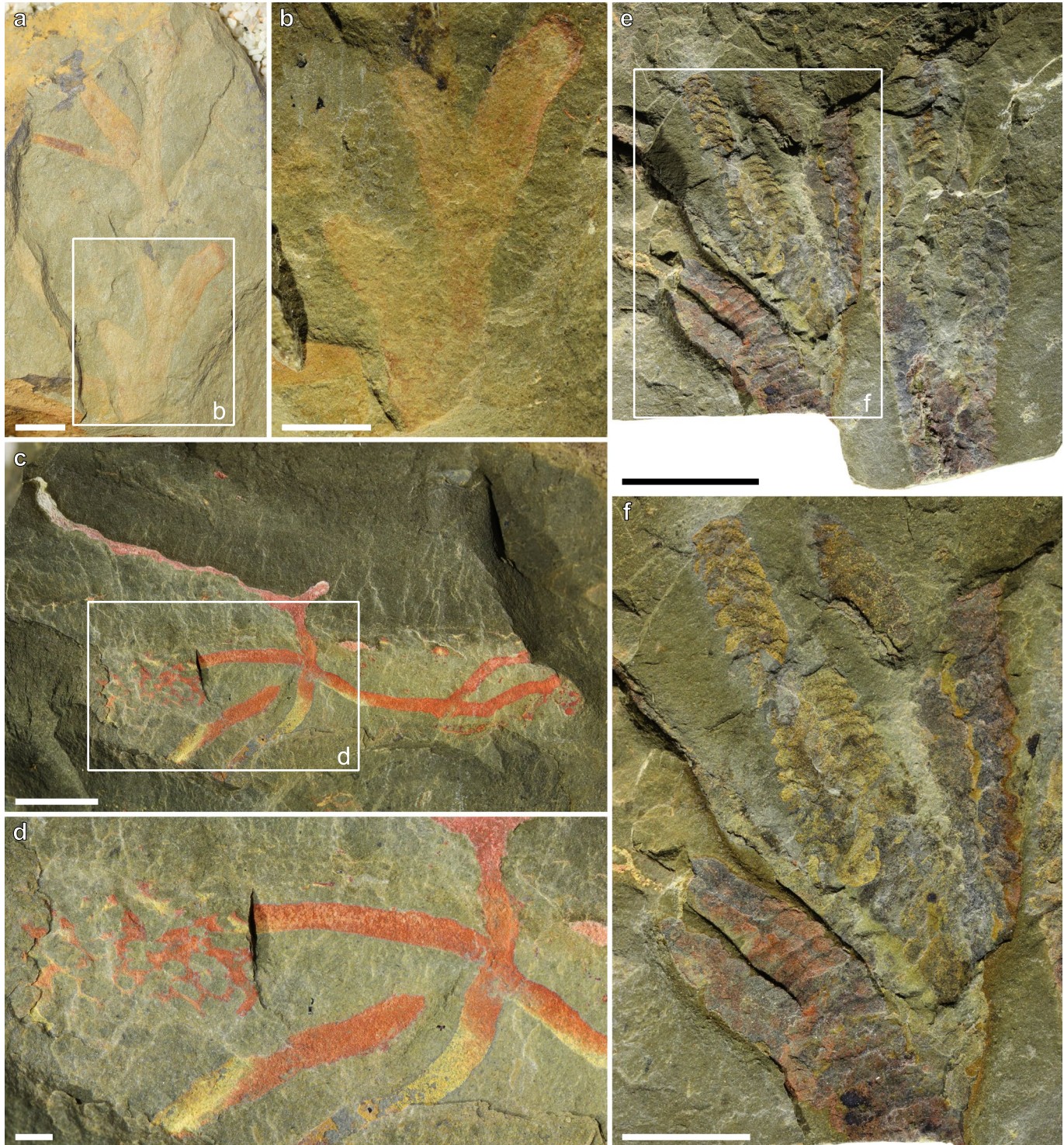

**Extended Data Fig. 7 | Additional views of the algae shown in Fig. 2. (a)**
Optical photograph of the thick branching algae UCBL-FSL713606 shown in
Fig. 2d. (**b**) Close-up view under polarised light of basal branches, from the
box in (a). (**c**) Optical photograph under polarised light of the thin branching
algae UCBL-FSL713607 shown in Fig. 2e. (**d**) Close-up view under polarised light
from the box in (c) highlighting textural differences between algae (right) and
sponges (left). (**e**) Optical photograph under low angle and polarised light of the
more complex algae UCBL-FSL713608 shown in Fig. 2f. (**f**) Close-up view under
polarised light from the box in (e) highlighting branched clumps of flat kidney-
shaped segments. Scale bars represent 5 cm in (a), (b), (c) and (f), 1 mm in (d),
and 1 cm in (e).

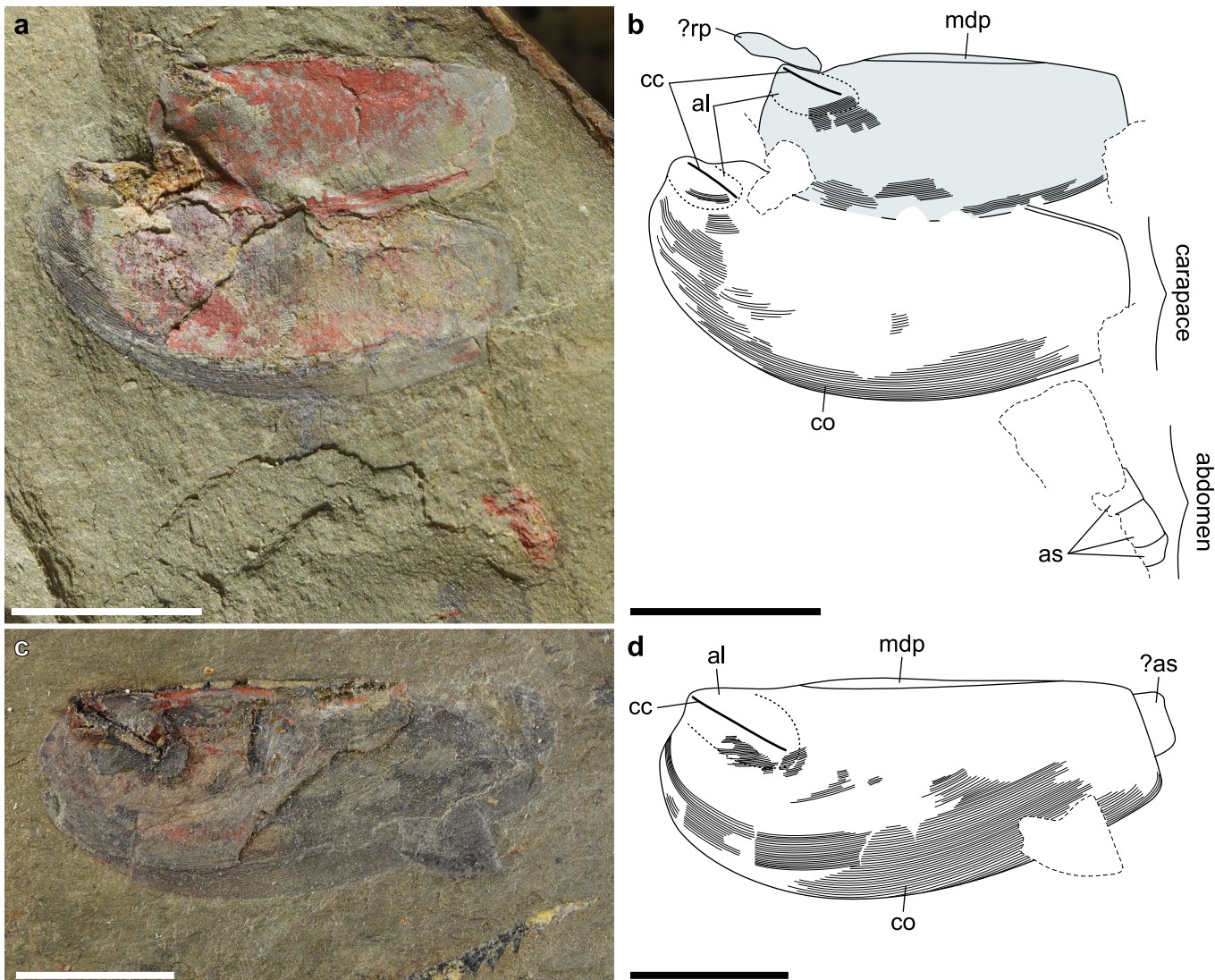

**Extended Data Fig. 8 | Phyllocarid crustaceans from the Cabrières Biota.**
(**a**, **b**) Optical photograph (a) and interpretative line drawing (b) of the counterpart of UCBL-FSL713609 shown in Fig. 3a. The specimen contains two left valves ornamented with very closely spaced, longitudinal striations. The bottom valve (white) is associated with a few abdominal segments. Fossil remains located anteriorly to the top valve (light blue) may represent a rostral plate. (**c**, **d**) Optical photograph (c) and interpretative line drawing (d) of the left valve UCBL-FSL713610 shown in Fig. 3b. Abbreviations: al, anterodorsal lobe; as, abdominal segments; cc, cephalic carina; co, carapace ornamentation; mdp, median dorsal plate; rp, rostral plate. Scale bars represent 5 mm.

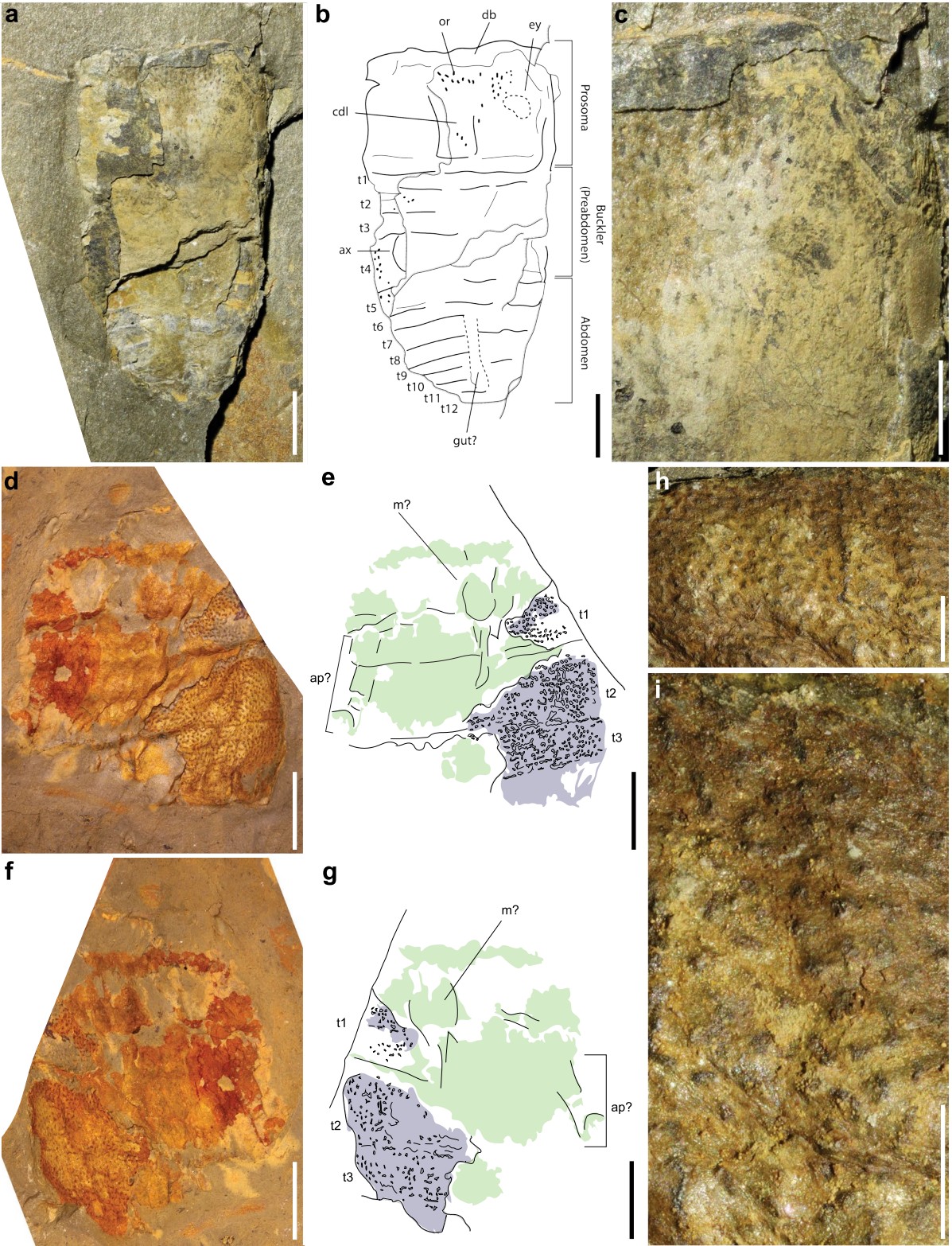

**Extended Data Fig. 9 | Additional images and interpretative drawings of the chelicerates in Fig. 3e, f.** (**a**, **b**) Optical photograph (a) and interpretative line drawing (b) of the chelicerate UCBL-FSL713613. The presence of a prosoma lacking enlarged axial nodes on opisthosomal tergites, reduced genal spines, and an opisthosoma that is largest at the third and fourth tergite, likely suggest chasmataspid affinities. (**c**) Close up of the eye and the ornamentation. (**d**–**g**) Optical photographs under polarised light of the part (d) and the counterpart (f), and corresponding interpretative line drawings (e, g) of the chelicerate UCBL-FSL713614. Green highlights ventral features. Pale purple shows dorsal features. The sample displays a carapace, three tergites, and a possible appendage. The presence of a possible metastoma suggest non-arachnid dekatriatan affinities. (**h**, **i**) Close-up photographs showing the ornamentation. Abbreviations: appendage (ap); (ax) axis; cardiac lobe (cdl); doublure (db); ornamentation (or); eye (ey); metastoma (m); and tergite (t). Scale bars represent 2 mm in (c)s and (i), 3 mm in (h), 5 mm in (a) and (b), and 8 mm in (d), (e), (f), and (g).

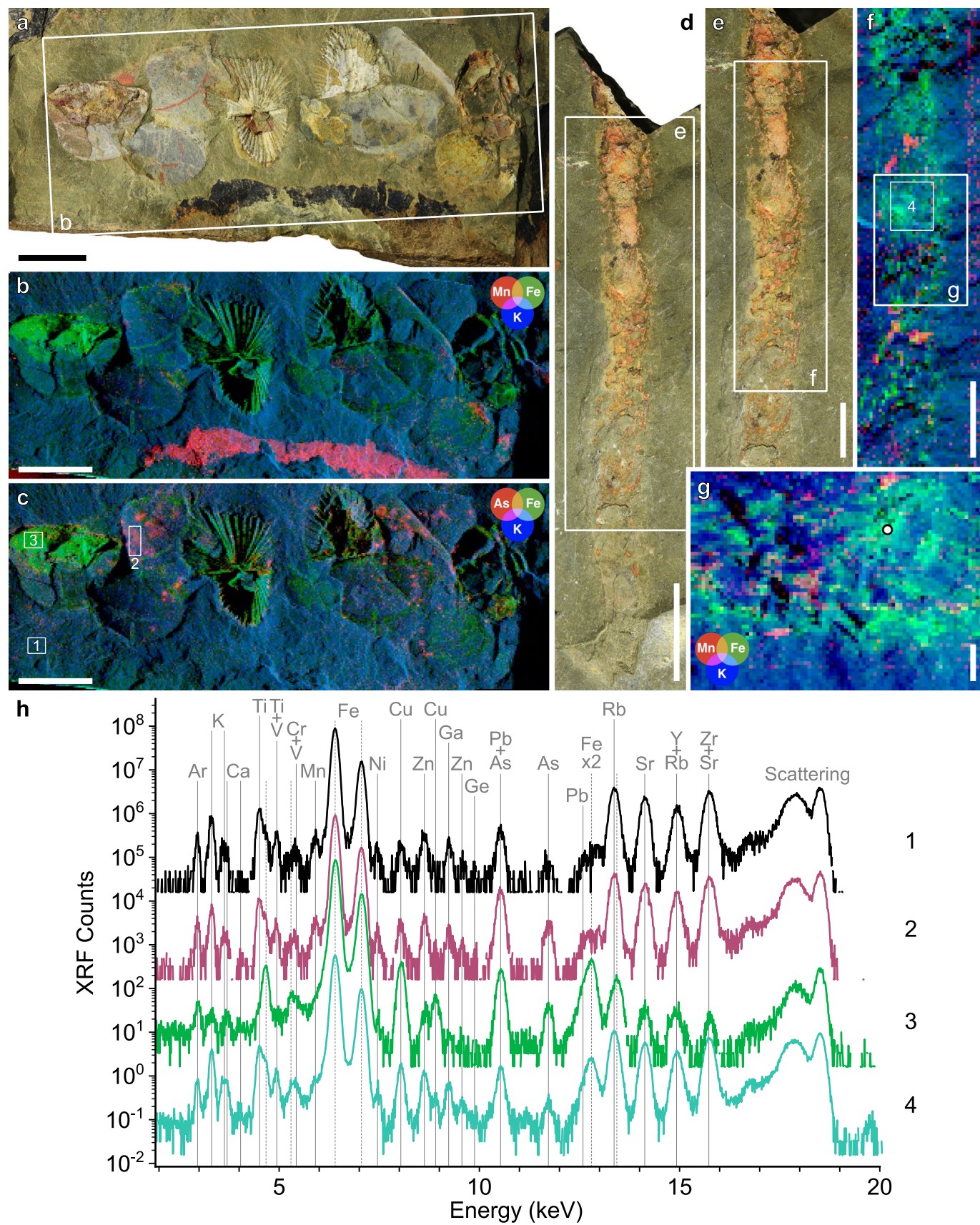

**Extended Data Fig. 10 | See next page for caption.**

**Extended Data Fig. 10 | Synchrotron-based X-ray fluorescence major-to-elemental mapping of the Cabrières Biota.** (**a**) Optical photograph under polarised light of the assemblage UCBL-FSL713602 shown in Fig. 1e. (**b, c**) False-colour overlays of manganese (red), iron (green) and potassium (blue) (b), and arsenic (red), iron (green) and potassium (blue) (c) distributions from the box in (a). Acquisition parameters: 200 $\mu$m steps, 80 ms dwell time, 51,100 pixels. (**d**) Optical photograph under polarised light of sponge UCBL-FSL713618. (**e**) Close-up view from the box in (d). (**f, g**) False-colour overlays of manganese (red), iron (green) and potassium (blue) distributions from the corresponding boxes in (e) and (f). Acquisition parameters: 200 $\mu$m steps, 60 ms dwell time, 6,150 pixels in (f); 100 $\mu$m steps, 60 ms dwell time, 4,800 pixels in (g). The white circle in (g) locates the XANES analysis shown in Fig. 4d. (**h**) Average μXRF spectra and main elemental contributions from the corresponding numbered boxes in c (156 pixels) and f (208 pixels). Scale bars represent 1 cm in (a), (b), (c) and (d), 5 mm in (e) and (f), and 1 mm in (g).

# Reporting Summary

## Statistics

For all statistical analyses, confirm that the following items are present in the figure legend, table legend, main text, or Methods section.

| n/a | Confirmed | |
|---|---|---|
| ☒ | ☐ | The exact sample size (*n*) for each experimental group/condition, given as a discrete number and unit of measurement |
| ☒ | ☐ | A statement on whether measurements were taken from distinct samples or whether the same sample was measured repeatedly |
| ☒ | ☐ | The statistical test(s) used AND whether they are one- or two-sided<br>*Only common tests should be described solely by name; describe more complex techniques in the Methods section.* |
| ☒ | ☐ | A description of all covariates tested |
| ☒ | ☐ | A description of any assumptions or corrections, such as tests of normality and adjustment for multiple comparisons |
| ☒ | ☐ | A full description of the statistical parameters including central tendency (e.g. means) or other basic estimates (e.g. regression coefficient) AND variation (e.g. standard deviation) or associated estimates of uncertainty (e.g. confidence intervals) |
| ☒ | ☐ | For null hypothesis testing, the test statistic (e.g. $F$, $t$, $r$) with confidence intervals, effect sizes, degrees of freedom and $P$ value noted<br>*Give P values as exact values whenever suitable.* |
| ☒ | ☐ | For Bayesian analysis, information on the choice of priors and Markov chain Monte Carlo settings |
| ☒ | ☐ | For hierarchical and complex designs, identification of the appropriate level for tests and full reporting of outcomes |
| ☒ | ☐ | Estimates of effect sizes (e.g. Cohen's *d*, Pearson's *r*), indicating how they were calculated |

*Our web collection on statistics for biologists contains articles on many of the points above.*

## Software and code

Policy information about availability of computer code

| Data collection | The following softwares were used during electron microscopy, mutispectral imaging, and optical photographs data collections: Helicon Focus software, ImageJ, Bruker Nano Analytics and PyMCA. |
|---|---|
| Data analysis | No software or code was used for data analysis. |

For manuscripts utilizing custom algorithms or software that are central to the research but not yet described in published literature, software must be made available to editors and reviewers. We strongly encourage code deposition in a community repository (e.g. GitHub). See the Nature Portfolio guidelines for submitting code & software for further information.

## Data

Policy information about availability of data

All manuscripts must include a data availability statement. This statement should provide the following information, where applicable:

- Accession codes, unique identifiers, or web links for publicly available datasets
- A description of any restrictions on data availability
- For clinical datasets or third party data, please ensure that the statement adheres to our policy

| All the data needed to reproduce this manuscript are available in the Main Text and the Extended Data Figures. All material can be accessed at the public collections of the University of Lyon1 (France). The collections are public with no access restrictions. All necessary information have been deposited with the specimens at the collections. |
|---|

# Research involving human participants, their data, or biological material

Policy information about studies with human participants or human data. See also policy information about sex, gender (identity/presentation), and sexual orientation and race, ethnicity and racism.

| | |
|---|---|
| Reporting on sex and gender | n/a |
| Reporting on race, ethnicity, or other socially relevant groupings | n/a |
| Population characteristics | n/a |
| Recruitment | n/a |
| Ethics oversight | n/a |

Note that full information on the approval of the study protocol must also be provided in the manuscript.

# Field-specific reporting

Please select the one below that is the best fit for your research. If you are not sure, read the appropriate sections before making your selection.

☐ Life sciences    ☐ Behavioural & social sciences    ☒ Ecological, evolutionary & environmental sciences

For a reference copy of the document with all sections, see nature.com/documents/nr-reporting-summary-flat.pdf

# Ecological, evolutionary & environmental sciences study design

All studies must disclose on these points even when the disclosure is negative.

| | |
|---|---|
| Study description | This study describes new fossil assemblage from the Early Ordovician of France |
| Research sample | Fossils found at outcrops were described to give a view of the general biota |
| Sampling strategy | Palaeontological excavation of an outcrop using classical tools such as hammers |
| Data collection | The authors collected all the fossils found at outcrop, and inferred their taxonomic position |
| Timing and spatial scale | Fieldwork happened over the past 5 years whenever weather permitted, with a total time spent on outcrop of over 50 days. Fossils come from a single quarry with a 1m thickness |
| Data exclusions | No data is excluded as all recognizable fossils were collected from the outcrop |
| Reproducibility | The fossils are deposited in a public museum and can be accessed by anyone who wishes to restudy these |
| Randomization | n/a |
| Blinding | n/a |

Did the study involve field work?    ☒ Yes    ☐ No

# Field work, collection and transport

| | |
|---|---|
| Field conditions | Fieldwork happened in spring and summer |
| Location | Cabrières Village Southern France, details are provided in the manuscript text, and with the fossils in the collections |
| Access & import/export | All data collection happened in accordance with local and French law. All fossils are housed in a public institution in France, their country of origin. |
| Disturbance | Only small tools, like hammers, were used to excavate the site. No major disturbances to surrounding fauna or flora happened when excavating the fossils. |

# Reporting for specific materials, systems and methods

We require information from authors about some types of materials, experimental systems and methods used in many studies. Here, indicate whether each material, system or method listed is relevant to your study. If you are not sure if a list item applies to your research, read the appropriate section before selecting a response.

## Materials & experimental systems

| n/a | Involved in the study |
|-----|----------------------|
| ☒ ☐ | Antibodies |
| ☒ ☐ | Eukaryotic cell lines |
| ☐ ☒ | Palaeontology and archaeology |
| ☒ ☐ | Animals and other organisms |
| ☒ ☐ | Clinical data |
| ☒ ☐ | Dual use research of concern |
| ☒ ☐ | Plants |

## Methods

| n/a | Involved in the study |
|-----|----------------------|
| ☒ ☐ | ChIP-seq |
| ☒ ☐ | Flow cytometry |
| ☒ ☐ | MRI-based neuroimaging |

## Palaeontology and Archaeology

| | |
|---|---|
| Specimen provenance | From outcrops near the Cabrières Village in France. No special permit is needed for this site at the moment. However, fieldwork permit were obtained from the University of Lyon1 when researchers visited the site. Also all discovered fossils are permanently housed in their country of origin, France. |
| Specimen deposition | Specimens are deposited in the public collections of the University Claude Bernard, Lyon1 in France |
| Dating methods | Widely accepted graptolite and trilobite biozones dating the outcrop to the Early Ordovician |

☒ Tick this box to confirm that the raw and calibrated dates are available in the paper or in Supplementary Information.

| | |
|---|---|
| Ethics oversight | No ethical approval was needed. |

Note that full information on the approval of the study protocol must also be provided in the manuscript.

## Plants

| | |
|---|---|
| Seed stocks | n/a |
| Novel plant genotypes | n/a |
| Authentication | n/a |

