## [Peer Review File · Nature Ecology & Evolution]

Peer Review Information

Journal: Nature Ecology & Evolution

Manuscript Title: The Cabrières Biota (France) provides insights into Ordovician polar ecosystems

Corresponding author name(s): Farid Saleh

Editorial Notes:

Reviewer Comments & Decisions:

Decision Letter, initial version:

2nd October 2023

Dear Farid,

It was nice to meet you at PalAss, and I'm glad that I'm now writing to you (finally) with reports on your manuscript entitled "The Cabrières Biota (France) provides insights into Ordovician polar ecosystems". The manuscript has been seen by three reviewers, and while they're enthusiastic about the contribution, they feel that the presentation, taxonomy and imaging still needs further work to become publishable. These concerns are laid out fairly clearly in the reports, I think, but do let me know if there's anything you'd like to discuss. It would also be helpful if you could let me know how long the required changes might take to make, should you feel able to undertake them.

We will therefore need to see your responses to the criticisms raised and to some editorial concerns, along with a revised manuscript, before we can reach a final decision regarding publication.

We therefore invite you to revise your manuscript taking into account all reviewer and editor comments. Please highlight all changes in the manuscript text file.

* If you have not done so already please begin to revise your manuscript so that it conforms to our Article format instructions at <http://www.nature.com/natecolevol/info/final-submission>. Refer also to any guidelines provided in this letter.

* Include a revised version of any required reporting checklist. It will be available to referees (and, potentially, statisticians) to aid in their evaluation if the manuscript goes back for peer review. A

2revised checklist is essential for re-review of the paper.

[REDACTED]

Nature Ecology & Evolution is committed to improving transparency in authorship. As part of our efforts in this direction, we are now requesting that all authors identified as 'corresponding author' on published papers create and link their Open Researcher and Contributor Identifier (ORCID) with their account on the Manuscript Tracking System (MTS), prior to acceptance. ORCID helps the scientific community achieve unambiguous attribution of all scholarly contributions. You can create and link your ORCID from the home page of the MTS by clicking on 'Modify my Springer Nature account'. For more information please visit www.springernature.com/orcid.

[REDACTED]

Reviewer expertise:

Reviewer #1: signed report

Reviewer #2: Ordovician Lagerstätte, including the Fezouata Biota

Reviewer #3: Cambrian and Ordovician Lagerstätte

Reviewers' comments:

Reviewer #1 (Remarks to the Author):

Re-Review Saleh et al. The Cabrières Biota (France) provides insights into Ordovician polar

2ecosystems

Dear Farid and colleagues,

This is a very interesting manuscript on a new soft-bodied fauna from the Ordovician polar region. The new deposit is well presented and the significance of its paleogeography is immense. The manuscript is also well written and increases our knowledge of the Ordovician diversity significantly. However, the fossil illustration leaves a bit to desire, especially as the extended data is not used to show the wealth of the diversity of this deposit, but rather shows the same fossils as in the main manuscript or their counterparts. Overall, I think it will be a great fit for Nature Eco Evo ones a few issues have been addressed.

All the best,
Julien Kimmig

Major comments:

It would be nice if the information from Extended Data Figure 2 would actually be in the text, as well as some quantification on how many taxa are soft-bodied or preserve soft-tissues. This is especially important as you call the new deposit a Konservat Lagerstätte.

Another important distinction to make is between exceptional preservation and the preservation of soft-tissues. Many of the soft-bodied fossils that are figured are not really exceptionally preserved, as they are not complete, i.e. the arthropods, or do not show the fine structures we know from other soft-tissue deposits. This does not discount the importance of this find, but I would like to see a more reserved approach to the use of exceptional.

Would it be possible to have a detailed stratigraphy of the outcrop/outcrops highlighting the fossil occurrences? If the data exists something similar to Yang et al. 2021 Nat Eco Evo Extended Data Figure 1 would be a great way to clarify what parts of the Landeyran Formation actually preserve the soft-bodied fossils.

Some minor comments:

Line 50-52: Kimmig et al. 2019 Science of Nature had an updated version of Van Roys Stratigraphic chart showing the occurrences of soft-bodied fossils in the Ordovician, might be worth adding. Also Moysiuk et al. 2022 J Pal.

Line 75 and after: biota/s preserving soft-bodied animals instead of exceptional biota/s

Line 156-167: Do you have any information on how thick the preservation horizon is, how far it extends, if there are different horizons preserving the soft-bodied fauna?

Line 173-186: Some sort of quantification would help the reader here.

3The main text figures are nice, but many of the indicated structures are difficult to see. This is especially the case in Figure 5, where many of the indicated structures are not easily identifiable. It would be good to increase the size of the individual pictures or add close-ups in the main text to better show these structures.

Reviewer #2 (Remarks to the Author):

Dear Authors,

This paper is clearly arranged, well written and well illustrated. The results are sufficiently backed with available data.

The article will make a good contribution to our evolving knowledge on such kind of preservation in two Lagerstätten : The Cabrières Biota and The Fezouata Biota.

Considered as an initial comparative reference, the Cabrières Biota showcases a notably higher abundance of algae and sponges, while echinoderms are remarkably scarcer in comparison to the Fezouata Biota. The other animal taxa display relatively comparable representation between the two Lagerstätten.

Similarities are also seen in the chemical signatures and such comparisons will enable a more comprehensive taphonomic analysis of the modes and mechanisms of preservation within other Konservat- Lagerstätten.

Finally, The Cabrières Biota is offering a unique opportunity to understand the ecology of past polar ecosystems.

Some very minor changes are made on the paper, see attached file

Reviewer #3 (Remarks to the Author):

The exceptional preserved fossil deposits in Early Palaeozoic provide a wealth of information on the evolution of past life and significantly enhancing our understanding of previous ecosystems. However, the sharp decline in these deposits after the Cambrian inferred to signify the closure of the early Phanerozoic taphonomic window. The discovery of Burgess Shale-type fossils within the Ordovician strata suggests that the Burgess Shale-type window remained open. In this study, the authors described the diversity and preservation of the Cabrières Biota, a newly discovered Early Ordovician Konservat-Lagerstätte from Montagne Noire, southern France. The Cabrières Biota show a diverse assemblage of both biomineralised and soft-bodied organisms predominantly preserved in iron oxides. The research of the Cabrières Biota will provide insights into Ordovician marine ecosystems and deserves to be published. The manuscript was well written, but it at the present failed to show the iconic taxa from the new deposit. Specifically, the algae, sponges, worms, branchiopods and skeleton of the arthropods recognized in this study are common among the Lagerstätten from the Cambrian to the Ordovician. The evidence of the abundance of sponges and the algae in the fossil assemblage is slightly insufficient to reveal the characteristics of the polar marine ecosystem. Therefore, I invite the

4authors to make moderate/major revisions before the article publishing.

Some of the taxonomic work in this research may be not very convincing. In the case the lobopodian fossil (Fig. 5h), the diagnostic feature of the Cambrian lobopodian is obscure. It is actually a bit difficult for me to recognize some of the anatomical details of the non-mineralized taxa described in this study.

The authors proposed that taxonomic composition of the Cabrières Biota is unique for the Ordovician deposits. The assemblage is dominated by algae and sponges, which was interpreted as specific characteristics and adaptations in polar settings. I am afraid I cannot fully agree with the authors on this view. Because a diverse sponge assemblage preserved with labile tissues was reported from the low latitude deposit, such as the Ordovician Anji Biota (Wu et al., 2022). Furthermore, the Fezouata Biota located at the Ordovician polar area does not show the same taxonomic composition to the Cabrières Biota.

It is generally believed that sponge animals are more tolerant of hypoxic environments, and the composition difference in various biotas was caused by the redox condition of water bodies. The further study of the geochemistry is required to determine the unique palaeoenvironment of the Cabrières deposit, which will help us to understand the taxonomic differences between the Cabrières biota and the contemporaneous exceptional preserved biotas. Alternatively, the limited number of specimens (ca. 400) might prevent from showing the accurate composition of the Cabrières biota. The fossil analyses in this study show that the organisms preserved in iron oxides, and the authors believe that preservation of the Cabrières Biota exhibits similarities with the preservation seen in the Fezouata Biota (Line 251), presenting the Burgess Shale-type exceptional preservation. It is reasonable to suppose that the original carbon was altered by the later intensive weathering in the Cabrières Biota. But the future collection of an expanded range of fossils from relatively fresh (low or moderately weathered) rock, will enable a more comprehensive taphonomic analysis of the modes and mechanisms of preservation within the Cabrières Biota and will facilitate comparisons with other early Palaeozoic Konservat-Lagerstätten.

*****END*****

Author Rebuttal to Initial comments

Kindly find below a detailed response to the comments raised by the three Reviewers. In the revised version we accounted for these comments by:

- Expanding the text
- Nuancing the writing
- Adding more quantifications
- Adding more arguments to the main conclusion
- Adding a chart of formations with soft-tissue preservation

5- Added a sedimentary log and outcrop photograph
- Enlarging previous photographs and using new images

We believe that we addressed the majority of comments, and we explained why certain points were not accounted for.

Reviewers' comments:

Reviewer #1 (Remarks to the Author):

Re-Review Saleh et al. The Cabrières Biota (France) provides insights into Ordovician polar ecosystems

Dear Farid and colleagues,

This is a very interesting manuscript on a new soft-bodied fauna from the Ordovician polar region. The new deposit is well presented and the significance of its paleogeography is immense. The manuscript is also well written and increases our knowledge of the Ordovician diversity significantly. However, the fossil illustration leaves a bit to desire, especially as the extended data is not used to show the wealth of the diversity of this deposit, but rather shows the same fossils as in the main manuscript or their counterparts. Overall, I think it will be a great fit for Nature Ecology and Evolution as a few issues have been addressed.

All the best,

Julien Kimmig

We would like to express our gratitude to Julien for his valuable comments, which have significantly contributed to the improvement of our manuscript. We have thoughtfully incorporated his suggestions into the main text and provided a detailed response below. It's important to clarify that the primary objective of this paper is to explore the broad spectrum of biodiversity at the new site and provide specific ecological interpretations related to its polar location. The supplementary data is used to present additional aspects of the paper that may be of interest to certain researchers, such as the specific details of the sponges and algae, as well as to highlight that some of the new arthropods belong to previously undescribed taxa. Consistent with common practice in our field, we have separated the sample photographs in the main text from the drawings, which are included in the supplementary material. This enhances the overall appeal of the main text to a wide readership. This approach aligns with the methods employed by other researchers (e.g., Van Roy et al. 2010, in Nature). It's also worth noting that in our paper, we present more than 20 fossils, which is standard for this type of research when discussing a new site. For comparison, Van Roy et al. in 2010, in Nature, presented approximately 20 specimens in their main manuscript and supplementary material. Similarly, Botting et al. in 2017, in their publication on the Anji Biota in Current Biology, included fewer than 15 specimens. Caron et al. in 2014, in Nature Communications, featured around 20 specimens in their study of the Marble Canyon. Fu et al. in 2019, in Science, showcased fewer than 30 specimens in their investigation of the Qingjiang Biota.

6Major comments:

It would be nice if the information from Extended Data Figure 2 would actually be in the text, as well as some quantification on how many taxa are soft-bodied or preserve soft-tissues. This is especially important as you call the new deposit a Konservat Lagerstätte.

We agree with Julien that information on abundance should be present in the main text. However, to facilitate visual comparison with the Fezouata Biota we opted for keeping the charts in the supplementary material and we added instead information on abundances whenever suitable in the text. For example: “The biota contains numerous taxa that exhibit biomineralisation (Fig. 3; Extended Data Fig. 2). These include animals such as molluscs, trilobites, brachiopods, hyoliths and cnidarians constituting respectively 14%, 12%, 9%, 7% and 6% of identified organisms in the Cabrières Biota (Extended Data Fig. 2). [...] In addition to trilobites, brachiopods, cnidarians, gastropods, and hyoliths, the Cabrières Biota is characterized by a prevalence of sponges and branching algae constituting 26% of all identified fossils (Extended Data Fig. 2). [...] Non-biomineralized arthropods form 16% of identified fossils found in the Cabrières Biota.” In terms of additional quantification, we added the following text to the paper: “Preliminary quantifications of the overall diversity within this Biota reveal that organisms with mineralized body walls (e.g., brachiopods, echinoderms, trilobites) that do not preserve soft tissues make up approximately 41% of the total diversity. In contrast, over half of the total diversity is comprised of non-biomineralized organisms, such as bivalved arthropods, chelicerates, lobopodians, hemichordates, or biomineralized animal groups that do preserve soft tissues, such the figured sponges (Fig. 4a, b). It is worth noting that these percentages are comparable to other well-known Lagerstätten from the Early Ordovician, such as the Fezouata Biota, which has approximately 44% of its taxa^{38, 39}, preserving only biomineralized remains.”

Another important distinction to make is between exceptional preservation and the preservation of soft-tissues. Many of the soft-bodied fossils that are figured are not really exceptionally preserved, as they are not complete, i.e. the arthropods, or do not show the fine structures we know from other soft-tissue deposits. This does not discount the importance of this find, but I would like to see a more reserved approach to the use of exceptional.

In the literature, the terms "exceptional preservation" and "preservation of soft tissues" are often used interchangeably, and there is no clear set of guidelines for the usage of these terms. However, to satisfy the comment raised by Julien, we have replaced “exceptional biota” with “biota preserving soft-bodied animals” whenever possible in the text (as suggested by a later comment from Julien). However, we would like to note that the previous usage of "exceptional preservation" in this paper was not arbitrary in the previous version of the text. This usage was based on the striking similarities between the Cabrières Biota and the Fezouata Shale, which is undeniably a site exhibiting both "exceptional preservation" and "preservation of soft tissues." In the past, the perception of the Fezouata Biota was influenced by specific collections in public institutions, primarily showcasing the "best" preserved organisms. However, fieldwork experience reveals that one may need to invest hours, and days, and even years, to discover a relatively complete sample or one with soft tissue preservation. A recent quantitative study, examining a level rich in stylophoran echinoderms from the Fezouata Biota (Saleh et al., in press), revealed that out

7of 423 stylophorans in that level, only one was fully articulated, while also providing a comprehensive view of the soft tissues within this animal group. It is important to note that fieldwork in the Fezouata Biota began in the early 2000s, which means more collection effort has been directed towards finding both “exceptional preservation” and the “preservation of soft tissues” in this site than in the Cabrières Biota. Nevertheless, despite this difference in collection efforts, the Cabrières Biota still managed to yield relatively complete organisms. The fidelity of preservation in many animal groups can be quite high as well, as exemplified by the longitudinal striations observed on the bivalved arthropod carapaces and the ornamentations on the chelicerates and the worms. In brief, we appreciate Julien's agreement with us regarding the significance of this discovery, and to eliminate any ambiguity on this matter we have also added the following text to the manuscript: “Many organisms in the Cabrières Biota can be fragmentary, which may indicate that they were either exposed to decay for relatively long periods of time or transported by sedimentary flows. Regardless of the processes responsible for such fragmentation, which will require further investigations, similar preservation is also observed in some localities from the Fezouata Biota, in which animals are dominantly fragmentary, with fully articulated organisms being the exception rather than the norm⁵⁰. Despite the difference in collection efforts between the Fezouata Biota and the Cabrières Biota, which was only recently discovered, the latter still yielded some complete organisms. Many animals are preserved in high detail as well, as exemplified by the longitudinal striations observed on the bivalved arthropod carapaces and the ornamentations on the chelicerates and the worms (Figs. 5a, b, e, f, 6 and see Extended Data Figs. 7, 8).”

Would it be possible to have a detailed stratigraphy of the outcrop/outcrops highlighting the fossil occurrences? If the data exists something similar to Yang et al. 2021 Nat Eco Evo Extended Data Figure 1 would be a great way to clarify what parts of the Landeyran Formation actually preserve the soft-bodied fossils.

A stratigraphic succession was added to Figure 2 in addition to a photograph of the most fossiliferous outcrop. The following text was added in the manuscript: “The biota is preserved in stratigraphically equivalent layers to the Landeyran Formation, but more to the east than traditional localities, specifically within the *Apatokephalus incisus* trilobite biozone²²⁻²⁵, which dates it to an upper Floian age²⁶ (Fl3; Fig. 2c; Extended Data Fig. 1). The Landeyran Formation corresponds to an offshore environment deposited in a transgressive phase^{27,28}, succeeding the sandy shoreface to upper offshore Foulon Formation^{25, 28}. However, a proper investigation based on recent knowledge of mud deposition²⁹ is needed to properly frame the sedimentary context. Soft-tissue preservation occurs within an interval of a one-meter thickness (Fig. 2d), located 15 meters above the base of the Landeyran formation (Fig. 2c).”

Some minor comments:

Line 50-52: Kimmig et al. 2019 Science of Nature had an updated version of Van Roys Stratigraphic chart showing the occurrences of soft-bodied fossils in the Ordovician, might be worth adding. Also Moysiuk et al. 2022 J Pal.

A stratigraphic table was added to Figure 1, and the references were added in the text.

Line 75 and after: biota/s preserving soft-bodied animals instead of exceptional biota/s

Done.

Line 156-167: Do you have any information on how thick the preservation horizon is, how far it extends, if there are different horizons preserving the soft-bodied fauna?

Detailed sedimentology, geochemistry, and stratigraphy will be the subject of further investigation. However, in response to Julien's comment, we have included a general stratigraphic log and photographs of the most fossiliferous outcrop in Figure 2. We have also added the following text in the manuscript: "Soft-tissue preservation occurs within an interval of a one-meter thickness (Fig. 2d), located 15 meters above the base of the Landeyran formation (Fig. 2c)."

Line 173-186: Some sort of quantification would help the reader here.

We added numerous quantifications as indicated in the answer to the first major comment above.

The main text figures are nice, but many of the indicated structures are difficult to see. This is especially the case in Figure 5, where many of the indicated structures are not easily identifiable. It would be good to increase the size of the individual pictures or add close-ups in the main text to better show these structures.

Figure 5 has now been edited and divided into two figures to be able to increase the size of individual panels. And we added new photographs for the lobopodians, using a new imaging technique (see Material and Methods).

Reviewer #2 (Remarks to the Author):

Dear Authors,

This paper is clearly arranged, well written and well illustrated. The results are sufficiently backed with available data.

The article will make a good contribution to our evolving knowledge on such kind of preservation in two Lagerstätten: The Cabrières Biota and The Fezouata Biota. Considered as an initial comparative reference, the Cabrières Biota showcases a notably higher abundance of algae and sponges, while echinoderms are remarkably scarcer in comparison to the Fezouata Biota. The other animal taxa display relatively comparable representation between the two Lagerstätten. Similarities are also seen in the chemical signatures and such comparisons will enable a more comprehensive taphonomic analysis of the modes and mechanisms of preservation within other Konservat-

9Lagerstätten. Finally, The Cabrières Biota is offering a unique opportunity to understand the ecology of past polar ecosystems. Some very minor changes are made on the paper, see attached file

We would like to thank Reviewer 2 for their comments. We have accounted for edits suggested by the Reviewer.

Reviewer #3 (Remarks to the Author):

The exceptional preserved fossil deposits in Early Palaeozoic provide a wealth of information on the evolution of past life and significantly enhancing our understanding of previous ecosystems. However, the sharp decline in these deposits after the Cambrian inferred to signify the closure of the early Phanerozoic taphonomic window. The discovery of Burgess Shale-type fossils within the Ordovician strata suggests that the Burgess Shale-type window remained open. In this study, the authors described the diversity and preservation of the Cabrières Biota, a newly discovered Early Ordovician Konservat-Lagerstätte from Montagne Noire, southern France. The Cabrières Biota show a diverse assemblage of both biomineralised and soft-bodied organisms predominantly preserved in iron oxides. The research of the Cabrières Biota will provide insights into Ordovician marine ecosystems and deserves to be published.

We would like to thank Reviewer 3 for his comments that helped us improve the manuscript. We have addressed his comments in the main text and in the detailed response below.

The manuscript was well written, but it at the present failed to show the iconic taxa from the new deposit. Specifically, the algae, sponges, worms, branchiopods and skeleton of the arthropods recognized in this study are common among the Lagerstätten from the Cambrian to the Ordovician.

We concur with Reviewer 3 that algae, sponges, worms, and many other groups are common among Cambrian and Ordovician Lagerstätten. Nowhere in the manuscript did we argue otherwise. What sets this paper apart is the unique taxonomic diversity observed in the Cabrières Biota, particularly when compared to other Early Ordovician Lagerstätten. This provides a novel perspective on the transition from the Cambrian Explosion to the Ordovician Radiation. This aspect was discussed in the previous version of the paper, where we made it clear that the Cabrières Biota distinguishes itself from the Fexouata Biota and the Liexi fauna, by the scarcity of echinoderms. This scarcity in echinoderms aligns more with the Fenxiang Biota and the Klabava Biota. Additionally, the Cabrières Biota is notable for its abundance of algae and sponges, similar to the Afon Gam Biota, but it stands out with a higher number of non-biomineralized arthropods. In the current version of the text, we further emphasize the distinctiveness of the Cabrières Biota, underlining more the disparities with the Fenxiang Biota and the Klabava Biota. The Cabrières Biota boasts a significantly higher population of arthropods compared to the Fenxiang Biota and lacks evidence of nematodes, scalidophorans, and corals. Furthermore, there are no bryozoans present in the Cabrières Biota in contrast to the Klabava Biota. In order to make this point clearer, this part was expanded upon in the current version of the text: “All animal groups in the Cabrières Biota are known from other Cambrian and Ordovician Lagerstätten, yet the taxonomic composition of the Cabrières Biota is particularly unique for the Early Ordovician. The newly described biota is almost as diverse as the range of clades seen in the Liexi Fauna⁵ and

10Fezouata Biota¹⁵, yet echinoderms, which are otherwise abundant in the Ordovician, are extremely rare in this biota (Fig. 2). This scarcity of echinoderms in the Cabrières Biota is similar to the Fenxiang Biota⁷ and the Klabava Biota¹⁶ but differs from other Early Ordovician Lagerstätten such as the Leixi Fauna and particularly the Fezouata Shale⁵¹⁻⁵⁷ (Extended Data Fig. 2). The Cabrières Biota yields a significantly higher population of arthropods compared to the Fenxiang Biota and lacks evidence of nematodes, scalidophorans, and corals. Furthermore, there are no bryozoans present in the Cabrières Biota in contrast to the Klabava Biota. The Cabrières Biota preserves an abundance of algae and sponges (Fig. 4a–f; Extended Data Fig. 2), similar to the Afon Gam Biota⁹, but with a greater number of non-biomineralised arthropods (Fig. 5a–f).”

The evidence of the abundance of sponges and the algae in the fossil assemblage is slightly insufficient to reveal the characteristics of the polar marine ecosystem. Therefore, I invite the authors to make moderate/major revisions before the article publishing.

The similarity between the Cabrières Biota and modern polar ecosystems is not solely based on the abundance of sponges and algae. This is one of several arguments previously presented in the paper. For instance, the Cabrières Biota's proximity to the Ordovician South Pole, the preservation of a diverse assemblage, the habitat selectivity of echinoderms, and other observed patterns all lend support to the notion that this assemblage represents a polar ecosystem characteristic of the Ordovician era. Each of these arguments was elaborated upon in the section titled "Ecological and Evolutionary Implications." In the revised manuscript, we have also introduced an additional argument—the absence of bryozoans, which are commonly found in other Ordovician Lagerstätten, and in modern tropical/temperate waters. The current summary text of this whole section reads as follow: “The position of the Cabrières Biota in the polar zone, the preservation of a diverse assemblage, the dominance of sponges and algae, and the habitat selectivity of echinoderms, among other observed patterns such as the absence of bryozoans which are dominantly found in tropical and temperate waters, all support the notion that this assemblage represents a polar ecosystem characteristic of the Early Ordovician.”

Some of the taxonomic work in this research may be not very convincing. In the case the lobopodian fossil (Fig. 5h), the diagnostic feature of the Cambrian lobopodian is obscure. It is actually a bit difficult for me to recognize some of the anatomical details of the non-mineralized taxa described in this study.

As per the recommendation of Reviewers 1 and 3, Figure 5 has been divided into two separate figures to increase the size of individual panels. Additionally, new photographs of the lobopodian fossils are now used to enhance the visibility of their distinctive features using a new imaging technique. We have also included the following new text regarding the lobopodians in the manuscript: “Some vermiform organisms are also present in the Cabrières Biota (~1% of identified fossils), one of which exhibits external ornamentation consisting of many tiny nodes and preserves gut remains (Fig. 6a). Two other specimens consist of a partially preserved elongated and annulated soft body bearing two thick oval plates (Fig. 6b, c). These plates are ca. 2 and 6 mm long in the first and second specimens, respectively, and present a complex internal morphology (Fig. 6b, c) with an outer surface displaying some reticulate ornamentation in places where thickness is preserved (Fig. 6d, e). A lateral extension at the base of one of the plates in the first specimen likely represents the remains of the proximal part of an appendage (?lo; Fig. 6b). At a similar position in the second specimen, a strong annulated area ends laterally into a series of lateral

11outgrowths (Fig. 6d, e) that likely represent spines or appendicules. The combination of a soft annulated body (and potentially appendages) and sclerite plates is characteristic of armoured lobopodians.”

The authors proposed that taxonomic composition of the Cabrières Biota is unique for the Ordovician deposits. The assemblage is dominated by algae and sponges, which was interpreted as specific characteristics and adaptations in polar settings. I am afraid I cannot fully agree with the authors on this view. Because a diverse sponge assemblage preserved with labile tissues was reported from the low latitude deposit, such as the Ordovician Anji Biota (Wu et al., 2022). Furthermore, the Fezouata Biota located at the Ordovician polar area does not show the same taxonomic composition to the Cabrières Biota.

Our initial objective was to focus mainly on other Early Ordovician sites. However, it is worth acknowledging that despite the Anji Biota's late Ordovician age (Botting et al., 2017), the substantial presence of sponges in this biota warrants mention in this manuscript. While both the Anji Biota and the Cabrières Biota exhibit a high abundance of sponges, their diversities vary. The Anji Biota, for instance, has yielded approximately 2,000 sponges, along with an abundance of graptolites, a single well-preserved articulated eurypterid with appendages, four nautiloid shells, and various rare mollusks, including gastropods. Many faunal elements present in the Cabrières Biota are absent in the Anji Biota, such as brachiopods, algae, bivalved arthropods, lobopodians, hemichordates, hyoliths, and trilobites. Furthermore, it's noteworthy to consider that the cooling events during the Ordovician, culminating in the Hirnantian Glaciation, may imply that the Anji Biota was deposited in relatively cold waters despite its low latitude location. This cooling phenomenon happening in the Ordovician suggests that similar communities to those found in Early Ordovician polar ecosystems might be expected in low-latitude environments during the Late Ordovician, as exemplified by the Anji Biota. To address this, we have included the following text in our manuscript: “Considering the global cooling happening after the Early Ordovician, similar ecosystems to the high-latitude Cabrières Biota can be anticipated in low-latitude environments during the Late Ordovician. Further global-scale investigations are required to confirm this trend, despite the presence of local paleontological evidence suggesting the existence of low-latitude sponge-dominated Lagerstätten in the Late Ordovician of China⁹⁵. Yet, it is possible to suggest that refugial zones found at high latitudes in the warm Early Ordovician migrated to lower latitudes by the cold Late Ordovician.”

It is generally believed that sponge animals are more tolerant of hypoxic environments, and the composition difference in various biotas was caused by the redox condition of water bodies. The further study of the geochemistry is required to determine the unique palaeoenvironment of the Cabrières deposit, which will help us to understand the taxonomic differences between the Cabrières biota and the contemporaneous exceptional preserved biotas.

While it is indeed true that sponge animals exhibit a higher tolerance for hypoxic environments, this phenomenon is predominantly observed in cases where overall diversity is low. In other words, sponges tend to dominate ecosystems with diminished biodiversity. However, this situation does not apply to the Cabrières Biota, where a diverse range of organisms coexist, including brachiopods, trilobites, bivalved arthropods, lobopodians, worms, cnidarians, hyoliths, and molluscs. Therefore, despite the need for future geochemical investigations, we can confidently dismiss the hypothesis of a low-oxygen environment as an explanation for the Cabrières Biota. This is

12due to the presence of numerous other groups colonizing the ecosystem. To address this comment, we have included the following text in the manuscript: “The prevalence of sponges in the Cabrières Biota cannot be ascribed to environmental factors like oxygen depletion, even though sponges typically thrive in hypoxic environments. This is because hypoxic environments are characterized by low diversity, which is clearly not the case for the Cabrières Biota preserving a diverse array of organisms, including brachiopods, trilobites, bivalved arthropods, lobopodians, worms, cnidarians, hyoliths, and molluscs.”

Alternatively, the limited number of specimens (ca. 400) might prevent from showing the accurate composition of the Cabrières biota. The fossil analyses in this study show that the organisms preserved in iron oxides, and the authors believe that preservation of the Cabrières Biota exhibits similarities with the preservation seen in the Fezouata Biota (Line 251), presenting the Burgess Shale-type exceptional preservation. It is reasonable to suppose that the original carbon was altered by the later intensive weathering in the Cabrières Biota. But the future collection of an expanded range of fossils from relatively fresh (low or moderately weathered) rock, will enable a more comprehensive taphonomic analysis of the modes and mechanisms of preservation within the Cabrières Biota and will facilitate comparisons with other early Palaeozoic Konservat-Lagerstätten.

This was already acknowledged in the previous version of the text where we wrote: “The future collection of an expanded range of fossils will enable a more comprehensive taphonomic analysis of the modes and mechanisms of preservation within the Cabrières Biota and will facilitate comparisons with other Lagerstätten³⁸⁻⁴².”

Decision Letter, first revision:

7th December 2023

Dear Farid,

Thank you for submitting your revised manuscript "The Cabrières Biota (France) provides insights into Ordovician polar ecosystems" (NATECOLEVOL-23081876A). It has now been seen again by the original reviewers and their comments are below. The reviewers find that the paper has improved in revision, and therefore we'll be happy in principle to publish it in Nature Ecology & Evolution, pending minor revisions to comply with our editorial and formatting guidelines.

13[REDACTED]

Reviewer #1 (Remarks to the Author):

The Authors have addressed all of the reviewers comments and I recommend the paper for publication in teh current form.

Sincerely,
Julien Kimmig

Reviewer #3 (Remarks to the Author):

Thank you for all the response. I think the manuscript has been carefully revised and it worth to be published in NEE.

Our ref: NATECOLEVOL-23081876A

21st December 2023

Dear Dr. Saleh,

Thank you for your patience as we've prepared the guidelines for final submission of your Nature Ecology & Evolution manuscript, "The Cabrières Biota (France) provides insights into Ordovician polar ecosystems" (NATECOLEVOL-23081876A). Please carefully follow the step-by-step instructions provided in the attached file, and add a response in each row of the table to indicate the changes that you have made. Please also check and comment on any additional marked-up edits we have proposed within the text. Ensuring that each point is addressed will help to ensure that your revised manuscript can be swiftly handed over to our production team.

****We would like to start working on your revised paper, with all of the requested files and forms, as soon as possible (preferably within two weeks). Please get in contact with us immediately if you anticipate it taking more than two weeks to submit these revised files.****

14When you upload your final materials, please include a point-by-point response to any remaining reviewer comments.

In recognition of the time and expertise our reviewers provide to Nature Ecology & Evolution's editorial process, we would like to formally acknowledge their contribution to the external peer review of your manuscript entitled "The Cabrières Biota (France) provides insights into Ordovician polar ecosystems". For those reviewers who give their assent, we will be publishing their names alongside the published article.

Nature Ecology & Evolution offers a Transparent Peer Review option for new original research manuscripts submitted after December 1st, 2019. As part of this initiative, we encourage our authors to support increased transparency into the peer review process by agreeing to have the reviewer comments, author rebuttal letters, and editorial decision letters published as a Supplementary item. When you submit your final files please clearly state in your cover letter whether or not you would like to participate in this initiative. Please note that failure to state your preference will result in delays in accepting your manuscript for publication.

Cover suggestions

We welcome submissions of artwork for consideration for our cover. For more information, please see our https://www.nature.com/documents/Nature_covers_author_guide.pdf guide for cover artwork.

Nature Ecology & Evolution has now transitioned to a unified Rights Collection system which will allow our Author Services team to quickly and easily collect the rights and permissions required to publish your work. Approximately 10 days after your paper is formally accepted, you will receive an email in providing you with a link to complete the grant of rights. If your paper is eligible for Open Access, our Author Services team will also be in touch regarding any additional information that may be required to arrange payment for your article.

Please note that *Nature Ecology & Evolution* is a Transformative Journal (TJ). Authors may publish their research with us through the traditional subscription access route or make their paper immediately open access through payment of an article-processing charge (APC). Authors will not be

15required to make a final decision about access to their article until it has been accepted. [Find out more about Transformative Journals](https://www.springernature.com/gp/open-research/transformative-journals)

Authors may need to take specific actions to achieve [compliance](https://www.springernature.com/gp/open-research/funding/policy-compliance-faqs) with funder and institutional open access mandates. If your research is supported by a funder that requires immediate open access (e.g. according to [Plan S principles](https://www.springernature.com/gp/open-research/plan-s-compliance)) then you should select the gold OA route, and we will direct you to the compliant route where possible. For authors selecting the subscription publication route, the journal's standard licensing terms will need to be accepted, including [self-archiving and license to publish](https://www.nature.com/nature-portfolio/editorial-policies/self-archiving-and-license-to-publish). Those licensing terms will supersede any other terms that the author or any third party may assert apply to any version of the manuscript.

[REDACTED]

[REDACTED]

Reviewer #1:

Remarks to the Author:

The Authors have addressed all of the reviewers comments and I recommend the paper for publication in teh current form.

Sincerely,

Julien Kimmig

Reviewer #3:

Remarks to the Author:

Thank you for all the response. I think the manuscript has been carefully revised and it worth to be published in NEE.

16Author Rebuttal, first revision:

Reviewer #1:

Remarks to the Author:

The Authors have addressed all of the reviewers comments and I recommend the paper for publication in the current form.

Sincerely,
Julien Kimmig

We thank Julien for his review and comments that helped us improve the manuscript.

Reviewer #3:

Remarks to the Author:

Thank you for all the responses. I think the manuscript has been carefully revised and it worth to be published in NEE.

We thank Reviewer 3 for their review and comments that helped us improve the manuscript.

Final Decision Letter:

10th January 2024

Dear Farid,

We are pleased to inform you that your Article entitled "The Cabrières Biota (France) provides insights into Ordovician polar ecosystems", has now been accepted for publication in Nature Ecology & Evolution.

17Over the next few weeks, your paper will be copyedited to ensure that it conforms to Nature Ecology and Evolution style. Once your paper is typeset, you will receive an email with a link to choose the appropriate publishing options for your paper and our Author Services team will be in touch regarding any additional information that may be required

Due to the importance of these deadlines, we ask you please us know now whether you will be difficult to contact over the next month. If this is the case, we ask you provide us with the contact information (email, phone and fax) of someone who will be able to check the proofs on your behalf, and who will be available to address any last-minute problems . Once your paper has been scheduled for online publication, the Nature press office will be in touch to confirm the details.

Acceptance of your manuscript is conditional on all authors' agreement with our publication policies (see www.nature.com/authors/policies/index.html). In particular your manuscript must not be published elsewhere and there must be no announcement of the work to any media outlet until the publication date (the day on which it is uploaded onto our web site).

Please note that *Nature Ecology & Evolution* is a Transformative Journal (TJ). Authors may publish their research with us through the traditional subscription access route or make their paper immediately open access through payment of an article-processing charge (APC). Authors will not be required to make a final decision about access to their article until it has been accepted. [Find out more about Transformative Journals](https://www.springernature.com/gp/open-research/transformative-journals)

Authors may need to take specific actions to achieve [compliance](https://www.springernature.com/gp/open-research/funding/policy-compliance-faqs) with funder and institutional open access mandates. If your research is supported by a funder that requires immediate open access (e.g. according to [Plan S principles](https://www.springernature.com/gp/open-research/plan-s-compliance)) then you should select the gold OA route, and we will direct you to the compliant route where possible. For authors selecting the subscription publication route, the journal's standard licensing terms will need to be accepted, including [self-archiving-and-license-to-publish](https://www.nature.com/nature-portfolio/editorial-policies/self-archiving-and-license-to-publish). Those licensing terms will supersede any other terms that the author or any third party may assert apply to any version of the manuscript.

We welcome the submission of potential cover material (including a short caption of around 40 words) related to your manuscript; suggestions should be sent to Nature Ecology & Evolution as electronic files (the image should be 300 dpi at 210 x 297 mm in either TIFF or JPEG format). Please note that such pictures should be selected more for their aesthetic appeal than for their scientific content, and that colour images work better than black and white or grayscale images. Please do not try to design a cover with the Nature Ecology & Evolution logo etc., and please do not submit composites of images related to your work. I am sure you will understand that we cannot make any promise as to whether any of your suggestions might be selected for the cover of the journal.

You can generate the link yourself when you receive your article DOI by entering it here: <http://authors.springernature.com/share>.

[REDATED]

P.S. Click on the following link if you would like to recommend Nature Ecology & Evolution to your librarian <http://www.nature.com/subscriptions/recommend.html#forms>

** Visit the Springer Nature Editorial and Publishing website at http://editorial-jobs.springernature.com?utm_source=ejp_NEcoE_email&utm_medium=ejp_NEcoE_email&utm_campaign=ejp_NEcoE for more information about our career opportunities. If you have any questions please click [here](mailto:editorial.publishing.jobs@springernature.com).**